# PEER LOSS FUNCTIONS: LEARNING FROM NOISY LABELS WITHOUT KNOWING NOISE RATES

## ABSTRACT

Learning with noisy labels is a common problem in supervised learning. Existing approaches require practitioners to specify *noise rates*, i.e., a set of parameters controlling the severity of label noises in the problem. The specifications are either assumed to be given or estimated using additional approaches. In this work, we introduce a technique to learn from noisy labels that does not require a priori specification of the noise rates. In particular, we introduce a new family of loss functions that we name as *peer loss* functions. Our approach then uses a standard empirical risk minimization (ERM) framework with peer loss functions. Peer loss functions associate each training sample with a certain form of "peer" samples, which evaluate a classifier' predictions jointly. We show that, under mild conditions, performing ERM with peer loss functions on the noisy dataset leads to the optimal or a near optimal classifier as if performing ERM over the clean training data, which we do not have access to. To our best knowledge, this is the first result on "learning with noisy labels without knowing noise rates" with theoretical guarantees. We pair our results with an extensive set of experiments, where we compare with state-of-the-art techniques of learning with noisy labels. Our results show that peer loss functions based method consistently outperforms the baseline benchmarks, as well as some recent new results. Peer loss provides a way to simplify model development when facing potentially noisy training labels, and can be promoted as a robust candidate loss function in such situations.

## 1 INTRODUCTION

The quality of supervised learning models depends on the training data $\{(x_n, y_n)\}_{n=1}^N$. In practice, label noise can arise due to a host of reasons. For instance, the observed labels $\tilde{y}_n$s may represent human observations of a ground truth label. In this case, human annotators may observe the label imperfectly due to differing degrees of expertise or measurement error, see e.g., medical examples such as labeling MRI images from patients. Many prior approaches to this problem in the machine learning literature aim to develop algorithms to learn models that are robust to label noise (Bylander, 1994; Cesa-Bianchi et al., 1999; 2011; Ben-David et al.; Scott et al., 2013; Natarajan et al., 2013; Scott, 2015). Typical approaches require *a priori* knowledge of *noise rates*, i.e., a set of parameters that control the severity of label noise. Working with unknown noise rates is difficult in practice: Often, one must estimate the noise rates from data, which may require additional data collection (Natarajan et al., 2013; Scott, 2015; Van Rooyen et al., 2015) (e.g., be a redundant set of noisy labels for each sample point, or a set of ground truth labels for tuning these parameters) and may introduce estimation error that can affect the final model in less predictable ways. Our main goal is to provide an alternative that does not require the specification of the noise rates, nor an additional estimation step for the noises. This target solution might help when the practitioner does not have access to reliable estimates of the noise rates (e.g., when the training data has limited size for the estimation tasks, or when the training data is already collected in a form that makes the estimation hard to perform).

In this paper, we introduce a new family of loss functions, *peer loss functions*, to empirical risk minimization (ERM), for a broad class of learning with noisy labels problems. Peer loss functions operate under different noise rates without requiring either *a priori* knowledge of the embedded noise rates, or an estimation procedure. This family of loss functions builds on approaches developed in the *peer prediction* literature (Miller et al., 2005; Dasgupta & Ghosh, 2013; Shnayder et al., 2016),

which studies how to elicit information from self-interested agents without verification. Typical approaches in the peer prediction literature design scoring functions to score each reported data using another noisy reference answer, without accessing ground truth information. We borrow this idea and the associated scoring functions via making a connection through treating each classifier's prediction as an agent's private information to be elicited and evaluated, and the noisy label as an imperfect reference from a "noisy label agent". The peer loss takes a form of evaluating classifiers' prediction using noisy labels on both the targeted samples and a particular form of constructed "peer" samples. The evaluation on the constructed peer sample encodes *implicitly* the information about the noises as well as the underlying true labels, which helps us offset the effects of label noises. The peer sample evaluation returns us a favorable property that expected risk of peer loss turns to be an affine transformation of the true risk of the classifier defined on the clean distribution. In other words, peer loss is invariant to label noises when optimizing with it. This effect helps us get rid of the estimation of noise rates.

The main contributions of this work are:

1. We propose a new family of loss functions that can easily adapt to existing ERM framework that i) is robust to *asymmetric* label noises with formal theoretical guarantees and ii) requires no prior knowledge or estimation of the noise rates (*no need for specifying noise rates*). We believe having the second feature above is a non-trivial progress, and it features a promising solution to deploy in an unknown noisy training environment.

2. We present formal results showing that performing ERM with a peer loss function can recover an optimal, or a near optimal classifier $f^*$ as if performing ERM on the clean data (Theorem 2, 3, 4). We also provide analysis for peer loss functions' risk guarantees (Theorem 5 and 7).

3. We present extensive experimental results to validate the usefulness of peer loss (Section 5 and Appendix). This result is encouraging as it is able to remove the long-standing requirement of learning error rates of noises (or estimating transition matrices as used in many relevant papers) before many of the existing methods can be applied. We also provide preliminary results on how peer loss generalizes to multi-class classification problems.

4. We will contribute to the community by publishing our codes and implementations.

## 1.1 RELATED WORK

**Learning from Noisy Labels** Our work fits within a stream of research on learning with noisy labels. A large stream of research on this topic works with the *random classification noise* (RCN) model, where observed labels are flipped independently with probability $e \in [0, \frac{1}{2}]$ (Bylander, 1994; Cesa-Bianchi et al., 1999; 2011; Ben-David et al.). Recently, learning with asymmetric noisy data (or also referred as *class-conditional* random classification noise (CCN)) for binary classification problems has been rigorously studied in (Stempfel & Ralaivola, 2009; Scott et al., 2013; Natarajan et al., 2013; Scott, 2015; Van Rooyen et al., 2015; Menon et al., 2015). For a more thorough survey of classical results on learning with noisy data, please refer to (Frénay & Verleysen, 2014).

**Symmetric loss** For RCN, where the noise parameters are symmetric, there exists works that show symmetric loss functions (Manwani & Sastry, 2013; Ghosh et al., 2015; 2017; Van Rooyen et al., 2015) are robust to the underlying noises, without specifying the noise rates. It was also shown that under certain conditions, the proposed loss functions are able to handle asymmetric noises. Our focus departs from this line of works, and we will exclusively focus on asymmetric noise setting, and study the possibility of an approach that can ignore the knowledge of noise rates.

Follow-up works (Du Plessis et al., 2013; van Rooyen et al., 2015; Menon et al., 2015; Charoenphakdee et al., 2019) have looked into leveraging symmetric conditions and 0-1 loss with asymmetric noises, and with more evaluation metrics, such as balanced error rate and AUROC. In particular, experimental evidences are reported in (Charoenphakdee et al., 2019) on the importance of symmetricity when learning with noisy labels.

**More recent works** More recent developments include an importance re-weighting algorithm (Liu & Tao, 2016), a noisy deep neural network learning setting (Sukhbaatar & Fergus, 2014; Han et al., 2018; Song et al., 2019), and learning from massive noisy data for image classification (Xiao et al., 2015), robust cross entropy loss for neural network (Zhang & Sabuncu, 2018), loss correction (Pa-

trini et al., 2017), among many others. Loss or sample correction has also been studied in the context of learning with unlabeled data with weak supervisions (Lu et al., 2018). Most of above works either lacks theoretical guarantee of the proposed method against asymmetric noise rates ((Sukhbaatar & Fergus, 2014; Zhang & Sabuncu, 2018)), or requires estimating the noise rate (or transition matrix between noisy and true labels, (Liu & Tao, 2016; Xiao et al., 2015; Patrini et al., 2017; Lu et al., 2018)). A good number of the recent works can be viewed as derivatives or extention of the unbiased surrogate loss function idea introduced in (Natarajan et al., 2013), therefore they would naturally require the knowledge of the noise rates or transition matrix. We do provide thorough comparisons between peer loss and the unbiased surrogate loss methods.

Mostly relevant to us is a recent work (Xu et al., 2019) that proposes an information theoretical loss (an idea adapted from an earlier theoretical contribution (Kong & Schoenebeck, 2018)) that is also robust to asymmetric noises rate. We aimed for a simple-to-optimize loss function that can easily adapt to existing ERM solutions. (Xu et al., 2019) involves estimating a joint distribution matrix between classifiers and noisy labels, and then invokes computing a certain information theoretical measure based on this matrix. Therefore, its sample complexity requirement and the sensitivity to noises in this estimation are not entirely clear to us (not provided in the paper either). We do provide calibration guarantees and generalization bounds. We provide conditions when the loss functions are convex. In general, we do think computationally peer loss functions are easy to optimize with, in comparing to information theoretical measures. Experiments comparing with (Xu et al., 2019) are also given in Section 5.

**Peer Prediction** Our work also builds on the literature for peer prediction (Prelec, 2004; Miller et al., 2005; Witkowski & Parkes, 2012; Radanovic & Faltings, 2013; Witkowski et al., 2013; Dasgupta & Ghosh, 2013; Shnayder et al., 2016; Liu & Chen, 2017). (Miller et al., 2005) established that strictly proper scoring rule (Gneiting & Raftery, 2007) could be adopted to elicit truthful reports from self-interested agents. Follow-up works that have been done to relax the assumptions imposed (Witkowski & Parkes, 2012; Radanovic & Faltings, 2013; Witkowski et al., 2013; Radanovic et al., 2016; Liu & Chen, 2017). Most relevant to us is (Dasgupta & Ghosh, 2013; Shnayder et al., 2016) where a correlated agreement (CA) type of mechanism was proposed. CA evaluates a report's correlations with another reference agent - its specific form inspired our peer loss.

## 2 PRELIMINARIES

*Notations and preliminaries:* For positive integer $n$, denote by $[n] := \{1, 2, ..., n\}$. Suppose $(X, Y) \in \mathcal{X} \times \mathcal{Y}$ are drawn from a joint distribution $\mathcal{D}$, with their marginal distributions denoted as $\mathbb{P}_X, \mathbb{P}_Y$ respectively. We assume $\mathcal{X} \subseteq \mathbb{R}^d$, and $\mathcal{Y} = \{-1, +1\}$, that is we consider a binary classification problem. Denote by $p := \mathbb{P}(Y = +1) \in (0, 1)$. There are $N$ training samples $(x_1, y_1), ..., (x_N, y_N)$ drawn i.i.d. from $\mathcal{D}$.

Instead of observing $y_n$s, the learner can only collect a noisy set of training labels $\tilde{y}_n$s, generated according to $y_n$s and a certain error rate model, that is we observe a dataset $\{(x_n, \tilde{y}_n)\}_{n=1}^N$. We assume a uniform error model for all the training samples we collect, in that errors in $\tilde{y}_n$s follow the same error rate model: denoting the random variable for noisy labels as $\tilde{Y}$ and we denote $e_{+1} := \mathbb{P}(\tilde{Y} = -1 | Y = +1)$, $e_{-1} := \mathbb{P}(\tilde{Y} = +1 | Y = -1)$ such that $0 \leq e_{+1} + e_{-1} < 1$. $e_{-1} + e_{+1} < 1$ is not unlike the condition imposed in the existing learning literature (Natarajan et al., 2013), and it simply implies that the noisy labels are positively correlating with the true labels (informative about the true labels). Label noises are conditional independent from the features, that is the error rate is uniform across $x_n$s: $\mathbb{P}(\tilde{Y} = y' | Y = y) = \mathbb{P}(\tilde{Y} = y' | X, Y = y), \forall y, y' \in \{-1, +1\}$. Denote the distribution of the noisy data $(X, \tilde{Y})$ as $\tilde{\mathcal{D}}$.

$f : \mathcal{X} \to \mathbb{R}$ is a real-valued decision function, and its risk w.r.t. the 0-1 loss is defined as $\mathbb{E}_{(X,Y) \sim \mathcal{D}}[\mathbb{1}(f(X), Y)]$. The Bayes optimal classifier $f^*$ is the one that minimizes the 0-1 risk: $f^* = \text{argmin}_f \mathbb{E}_{(X,Y) \sim \mathcal{D}}[\mathbb{1}(f(X), Y)]$. Denote this optimal risk as $R^*$. Instead of minimizing the above 0-1 risk, the learner often uses a surrogate loss function $\ell : \mathbb{R} \times \{-1, +1\} \to \mathbb{R}_+$, and find a $f \in \mathcal{F}$ that minimizes the following error: $\mathbb{E}_{(X,Y) \sim \mathcal{D}}[\ell(f(X), Y)]$. Denote the following measures:

$$R_{\mathcal{D}}(f) = \mathbb{E}_{(X,Y) \sim \mathcal{D}}[\mathbb{1}(f(X), Y)], \ R_{\ell, \mathcal{D}}(f) = \mathbb{E}_{(X,Y) \sim \mathcal{D}}[\ell(f(X), Y)].$$

When there is no confusion, we will also short-hand $\mathbb{E}_{(X,Y)\sim\mathcal{D}}[\ell(f(X),Y)]$ as $\mathbb{E}_{\mathcal{D}}[\ell(f(X),Y)]$. Using $D$ to denote a dataset collected from distribution $\mathcal{D}$ (correspondingly $\tilde{D} := \{(x_n, \tilde{y}_n)\}_{n=1}^N$ for $\tilde{\mathcal{D}}$), the empirical risk measure for $f$ is defined as $\hat{R}_{\ell,D}(f) = \frac{1}{|D|}\sum_{(x,y)\in D}\ell(f(x),y)$ .

## 2.1 LEARNING WITH NOISY LABELS

Typical methods for learning with noisy labels include developing bias removal surrogates loss function methods to learn with noisy data (Natarajan et al., 2013). For instance, Natarajan et al. (2013) tackle this problem by defining an *"un-biased" surrogate loss functions* over $\ell$ to help "remove" noise, when $e_{-1} + e_{+1} < 1$: $\tilde{\ell}(t,y) := \frac{(1-e_{-y})\cdot\ell(t,y)-e_y\cdot\ell(t,-y)}{1-e_{-1}-e_{+1}}, \forall t, y$. $\tilde{\ell}$ is identified such that when a prediction is evaluated against a noisy label using this surrogate loss function, the prediction is as if evaluated against the ground-truth label using $\ell$ in expectation. Hence the loss of the prediction is "unbiased", that is $\forall$ prediction $t$, $\mathbb{E}_{\tilde{Y}|y}[\tilde{\ell}(t,\tilde{Y})] = \ell(t,y)$ [Lemma 1, (Natarajan et al., 2013)].

One important note to make is most, if not all, existing solutions require the knowledge of error rates $e_{-1}, e_{+1}$. Previous works either assumed the knowledge of it, or needed additional clean labels or redundant noisy labels to estimate them. This becomes the bottleneck of applying these great techniques in practice. Our work is also motivated by the desire to remove this limitation.

## 2.2 PEER PREDICTION: INFORMATION ELICITATION WITHOUT VERIFICATION

Peer prediction is a technique developed to truthfully elicit information when there is no ground truth verification. Suppose we are interested in eliciting private observations about a binary event $y \in \{-1, +1\}$ generated according to a random variable $Y$. There are $K$ agents indexed by $[K]$. Each of them holds a noisy observation of $y$, denoted as $y(i) \in \{-1, +1\}$, $i \in [K]$. We would like to elicit the $y(i)$s, but they are completely private and we won't observe $y$ to evaluate agents' reports. Denote by $r(i)$ the reported data from each agent $i$. It is completely possible that $r(i) \neq y(i)$ if agents are not compensated properly for their information. Results in *peer prediction* have proposed scoring or reward functions that evaluate an agent's report using the reports of other peer agents. For example, a peer prediction mechanism may reward agent $i$ for her report $r(i)$ using $S(r(i), r(j))$ where $r(j)$ is the report of a randomly selected reference agent $j \in [K]\backslash\{i\}$. The scoring function $S$ is designed so that truth-telling is a strict Bayesian Nash Equilibrium (implying other agents truthfully report their $y(j)$s), that is, $\forall i\ \mathbb{E}_{y(j)}[S(y(i),y(j))|y(i)] > \mathbb{E}_{y(j)}[S(r(i),y(j))|y(i)], \forall r(i) \neq y(i)$.

**Correlated Agreement** (Shnayder et al., 2016; Dasgupta & Ghosh, 2013) (CA) is a recently established peer prediction mechanism for a multi-task setting [1]. CA is also the core and the focus of our subsequent sections on developing peer prediction based loss functions. This mechanism builds on a $\Delta$ matrix that captures the stochastic correlation between the two sources of predictions $y(i)$ and $y(j)$. Denote the following mapping function: $g(1) = -1, g(2) = +1$, $\Delta \in \mathbb{R}^{2\times 2}$ is then defined as a squared matrix with its entries defined as follows:

$$\Delta(k,l) = \mathbb{P}\big(y(i)=g(k), y(j)=g(l)\big) - \mathbb{P}\big(y(i)=g(k)\big)\mathbb{P}\big(y(j)=g(l)\big),\ k,l=1,2$$

The intuition of above $\Delta$ matrix is that each $(i,j)$ entry of $\Delta$ captures the marginal correlation between the two predictions. $M \in \mathbb{R}^{2\times 2}$ is defined as the sign matrix of $\Delta$: $M := \text{Sgn}(\Delta)$, where $\text{Sgn}(x) = 1, x > 0$; $\text{Sgn}(x) = 0$, o.w. Define the following score matrix

$$M^S : \{-1,+1\}\times\{-1,+1\} \rightarrow \{0,1\} : M^S(y,y') =: M(g^{-1}(y), g^{-1}(y')), \quad (1)$$

where $g^{-1}$ is the inverse function of $g$. CA requires each agent $i$ to perform multiple tasks: denote agent $i$'s observations for the $N$ tasks as $y_1(i), ..., y_N(i)$. Ultimately the scoring function $S(\cdot)$ for each task $k$ that is shared between $i, j$ is defined as follows: randomly draw two other tasks $k_1^p, k_2^p$,

$$S\big(y_k(i), y_k(j)\big) := M^S\big(y_k(i), y_k(j)\big) - M^S\big(y_{k_1^p}(i), y_{k_2^p}(j)\big),\ k_1^p \neq k_2^p \neq k$$

Note a key difference between the first and second $M^S$ terms is that the second term is defined for two independent peer tasks $k_1^p, k_2^p$ (as the reference answers). It was established in (Shnayder et al.,

---

[1] We provide other examples of peer prediction functions in the Appendix.

2016) that CA is truthful and proper (Theorem 5.2, Shnayder et al. (2016).) [2]; in particular, if $y(j)$ is *categorical* w.r.t. $y(i)$: $\mathbb{P}(y(j) = y'|y(i) = y) < \mathbb{P}(y(j) = y'), \forall i, j \in [K], \ y' \neq y$ then $S(\cdot)$ is strictly truthful (Theorem 4.4, Shnayder et al. (2016)).

## 3 LEARNING WITH NOISY DATA: A PEER PREDICTION APPROACH

In this section, we show that peer prediction scoring functions, when specified properly, will adopt Bayes optimal classifier as their maximizers (or minimizers for the corresponding loss form).

### 3.1 LEARNING WITH NOISY DATA AS AN ELICITATION PROBLEM

We first state our problem of learning with noisy labels as a peer prediction problem. The connection is made by firstly rephrasing the two data sources, the classifiers and the noisy labels, from agents' perspective. For a task $y \in \{-1, +1\}$, say $+1$ for example, denote the noisy labels $\tilde{Y}$ as $r(X), X \sim \mathbb{P}_{X|Y=1}$. In general, $r(X)$ can be interpreted as the agent that observes $\tilde{y}_1, ..., \tilde{y}_N$ for a set of randomly drawn feature vectors $x_1, ..., x_N$: $\tilde{y}_n \sim r(X)$. Suppose the agent's observations are defined as follows (similar to the definition of $e_{+1}, e_-$): $\mathbb{P}(r(X) = -1|Y = +1) = e_{+1}$, $\mathbb{P}(r(X) = +1|Y = -1) = e_{-1}$. Denote another agent whose observations "mimic" the Bayes optimal classifier $f^*$. Again denote this optimal classifier agent as $r^*(X) := f^*(X)$:

$$\mathbb{P}_X(r^*(X) = -1|Y = +1) = e^*_{+1}, \ \mathbb{P}_X(r^*(X) = +1|Y = -1) = e^*_{-1}$$

Suppose we would like to elicit predictions from the optimal classifier agent $r^*$, while the reports from the noisy label agent $r$ will serve as the reference reports. Both $r$ and $r^*$ are randomly assigned a task $x$, and each of them observes a signal $r(x)$ and $r^*(x)$ respectively. Denote the report from agent $r^*$ as $\tilde{r}^*$. A scoring function $S : \mathbb{R} \times \mathbb{R} \to \mathbb{R}$ is called to induce *strictly* truthfulness if the following fact holds: $\mathbb{E}_X\big[S\big(r^*(X), r(X)\big)\big] > \mathbb{E}_X\big[S\big(\tilde{r}^*, r(X)\big)\big], \forall \tilde{r}^* \neq r^*(X)$. Taking the negative of $S(\cdot)$ (changing a reward score one aims to maximize to a loss to minimize) we also

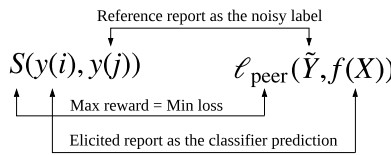

Figure 1: Illustration of our idea. $S$ is the peer prediction function; our $\ell_{\text{peer}}$ is to "evaluate" a classifier's prediction using a noisy reference.

have $\mathbb{E}_X\big[-S\big(r^*(X), r(X)\big)\big] < \mathbb{E}_X\big[-S\big(\tilde{r}^*, r(X)\big)\big], \forall \tilde{r}^* \neq r^*(X)$, implying when taking $-S(\cdot)$ as the loss function, minimizing $-S(\cdot)$ w.r.t. $R$ will return us the Bayes optimal classifier $f^*$. Our idea can be summarized easily using Fig. 1.

### 3.2 "PROPER" PEER PREDICTION FUNCTION INDUCED BAYES OPTIMAL CLASSIFIER

When there is no ambiguity, we will shorthand $r(X), r^*(X)$ as $r, r^*$, with keeping in mind that $r, r^*$ encode the randomness in $X$. Suppose $S(\cdot)$ is able to elicit the Bayes optimal classifier $f^*$ (agent $r^*$) using $r$, we have the following theorem formally:

**Theorem 1.** $f^* = argmin_f \ \mathbb{E}_{(X, \tilde{Y}) \sim \tilde{\mathcal{D}}}\big[-S(f(X), r)\big]$.

This proof can be done via showing that any non-optimal Bayes classifier corresponds to a misreporting strategy, thus establishing its non-optimality. We emphasize that it is not super restrictive to have a strictly truthful peer prediction scoring function $S$. We provide discussions in Appendix.

Theorem 1 provides a conceptual connection and can serve as an anchor point when connecting a peer prediction score function to the problem of learning with noisy labels. So far we have not discussed about a specific form of how we construct a loss function using ideas from peer prediction, and have not mentioned the requirement of knowing the noise rates. We will provide the detail about a particular *peer loss* in next section, and explain its independence of noise rates.

---

[2]To be precise, it is an informed truthfulness. We refer interested readers to (Shnayder et al., 2016) for the detailed differences.

## 4   PEER LOSS FUNCTION

We now present peer loss, a family of loss functions inspired by a particular peer prediction mechanism, the correlated agreement (CA), as presented in Section 2.2. We are going to show that peer loss is able to induce the minimizer of a concept class $\mathcal{F}$, under a broad set of non-restrictive conditions. In this Section, we do not restrict to Bayes optimal classifiers, nor do we impose any restrictions on the loss functions' elicitation power.

### 4.1   PREPARATION: EXPLAINING CA IN OUR CLASSIFICATION PROBLEM

To give a gentle start, we repeat the setting of CA for our classification problem.

**$\Delta$ and scoring matrix**   First recall that $\Delta \in \mathbb{R}^{2 \times 2}$ is a squared matrix with entries defined between $r^*$ (the $f^*$) and $r$ (i.e., the noisy labels $\tilde{Y}$):

$$\Delta(k,l) = \mathbb{P}\big(f^*(X) = g(k), \tilde{Y} = g(l)\big) - \mathbb{P}\big(f^*(x) = g(k)\big)\mathbb{P}\big(\tilde{Y} = g(l)\big), \ k, l = 1, 2$$

Recall $g(\cdot)$ is simply a mapping function: $g(1) = -1, g(2) = +1$. $\Delta$ characterizes the "marginal" correlations between the optimal classifier' prediction and the noisy label $\tilde{Y}$. Then the following scoring matrix $M \in \mathbb{R}^{n \times n}$, sign matrix of $\Delta$, $M := \text{Sgn}(\Delta)$ is computed.

**Example 1.** *Consider a binary class label case:* $\mathbb{P}(Y = -1) = 0.4, \mathbb{P}(Y = +1) = 0.6$, *the noises in the labels are* $e_{-1} = 0.3, e_{+1} = 0.4$ *and* $e^*_{-1} = 0.2, e^*_{+1} = 0.3$. *Then we have* $\Delta(1,1) = 0.036$, $\Delta(1,2) = -0.036$, $\Delta(2,1) = -0.036$, $\Delta(2,2) = 0.036$. *And:*
$$\Delta = \begin{bmatrix} 0.036 & -0.036 \\ -0.036 & 0.036 \end{bmatrix} \Rightarrow M = Sgn(\Delta) = \begin{bmatrix} 1 & 0 \\ 0 & 1 \end{bmatrix}.$$

**Peer samples**   For each sample $(x_i, \tilde{y}_i)$, randomly draw another two samples $(x_{i_1^p}, \tilde{y}_{i_1^p}), (x_{i_2^p}, \tilde{y}_{i_2^p})$ such that $i_1^p \neq i_2^p$ and $i_1^p, i_2^p \neq i$. We will name $(x_{i_1^p}, \tilde{y}_{i_1^p}), (x_{i_2^p}, \tilde{y}_{i_2^p})$ as $i$'s peer samples. After pairing $x_{i_1^p}$ with $\tilde{y}_{i_2^p}$ (two independent tasks), the scoring function $S(\cdot)$ for each sample point $x_i$ is defined as follows: $S(f(x_i), \tilde{y}_i) = M^S\big(f(x_i), \tilde{y}_i\big) - M^S\big(f(x_{i_1^p}), \tilde{y}_{i_2^p}\big)$. Recall $M^S(\cdot)$ is a sign score matrix defined for $\Delta$ (Eqn. (1)). Define loss function $\tilde{\ell}(\cdot)$ as the negative of $S(\cdot)$:

$$\text{(\textbf{Generic Peer Loss})} \ \ \tilde{\ell}\big(f(x_i), \tilde{y}_i\big) := \big(1 - M^S\big(f(x_i), \tilde{y}_i\big)\big) - \big(1 - M^S\big(f(x_{i_1^p}), \tilde{y}_{i_2^p}\big)\big). \quad (2)$$

The first term above evaluates the classifier's prediction on $x_i$ using noisy label $\tilde{y}_i$, and the second "peer" term defined on two independent tasks $i_1^p, i_2^p$ "punishes" the classifier from overly agreeing with the noisy labels. We will see this effect more clearly. According to Theorem 1, minimizing $\tilde{\ell}(\cdot)$ is going to find the Bayes optimal classifier, if $\tilde{Y}$ and $f^*$ are categorical, which is easily satisfied:

**Lemma 1.** *When* $e_{-1} + e_{+1} < 1$ *and* $e^*_{-1} + e^*_{+1} < 1$, $r$ *and* $r^*$ *($\tilde{Y}$ and $f^*$) are categorical.*

$e^*_{-1} + e^*_{+1} < 1$ means that the optimal classifier is at least informative ((Liu & Chen, 2017)) - if otherwise, we can flip the classifier's output to obtain one.

### 4.2   PEER LOSS

We need to know $\text{Sgn}(\Delta)$ in order to specify $M^S$ and $\tilde{\ell}$, which requires certain information about $f^*$ and $\tilde{Y}$. We show that for the cases that the literature is broadly interested in, $\text{Sgn}(\Delta)$ is simply the identify matrix (same condition as stated in Lemma 1):

**Lemma 2.** *If* $e_{-1} + e_{+1} < 1, e^*_{-1} + e^*_{+1} < 1$, *then* $Sgn(\Delta) = I_{2 \times 2}$, *i.e., the identity matrix.*

This is basically stating that for $\Delta(k,k), k = 1, 2$, $f^*$ and $\tilde{Y}$ are positively correlating, so the marginal correlation is positive; while for off-diagonal entries, they are negatively correlating.

**Peer loss**   When $\text{Sgn}(\Delta) = I_{2 \times 2}$, $M^S(y, y') = 1$ if $y = y'$, and 0 otherwise. $\tilde{\ell}(\cdot)$ defined in Eqn. (2) reduces to the following form:

$$\mathbb{1}_{\text{peer}}(f(x_i), \tilde{y}_i) = \mathbb{1}(f(x_i), \tilde{y}_i) - \mathbb{1}(f(x_{i_1^p}), \tilde{y}_{i_2^p}) \quad (3)$$

To see this, for instance $1 - M^S\big(f(x_i) = +1, \tilde{y}_i = +1\big) = 1 - M(2,2) = 1 - 1 = 0 = \mathbb{1}(f(x_i) = -1, \tilde{y}_i = +1)$. Replacing $\mathbb{1}(\cdot)$ with any generic loss $\ell(\cdot)$ we define:

$$(\textbf{Peer Loss}): \quad \ell_{\text{peer}}(f(x_i), \tilde{y}_i) = \ell(f(x_i), \tilde{y}_i) - \ell(f(x_{i_1^p}), \tilde{y}_{i_2^p}) \tag{4}$$

We name above loss as *peer loss*. This strikingly simple form of $\ell_{\text{peer}}(f(x_i), \tilde{y}_i)$ implies that knowing $e_{-1} + e_{+1} < 1, e_{-1}^* + e_{+1}^* < 1$ hold is all we need to specify $\ell_{\text{peer}}$.

**Why do we not need the knowledge of noise rates explicitly?** Both of the terms $\mathbb{1}(f(x_i), \tilde{y}_i)$ and $\mathbb{1}(f(x_{i_1^p}), \tilde{y}_{i_2^p})$ encoded the knowledge of noise rates *implicitly*. The carefully constructed form as presented in Eqn. 3 allows peer loss to be invariant against noises (Lemma 3, a property we will explain later). For a preview, for example if we take expectation of $\mathbb{1}_{\text{peer}}(f(x_i) = +1, \tilde{y}_i = +1)$ we will have $\mathbb{E}\left[\mathbb{1}_{\text{peer}}(f(x_i) = +1, \tilde{y}_i = +1)\right] = \mathbb{P}(f(X) = +1, \tilde{Y} = +1) - \mathbb{P}(f(X) = +1) \cdot \mathbb{P}(\tilde{Y} = +1)$, the marginal correlation between $f$ and $\tilde{Y}$, which is exactly capturing the entries of $\Delta$ defined between $f$ and $\tilde{Y}$! The second term above is a product of marginals because of the independence of peer samples $i_1^p, i_2^p$. Using the sign of $\Delta$ is all we need to recover this information measure in expectation. In other words, both the joint and marginal distribution terms encode the noise rate information in an implicit way. Later we will show this measure is invariant under label noises, which gives us the property of peer loss being invariant to label noises and the ability of dropping the requirement of knowing noise rates. We will instantiate this argument formally with Lemma 3 and establish a link between the above measure and the true risk of a classifier on the clean distribution. The rest of presentation focuses on $\ell_{\text{peer}}$ (Eqn. (4)), but $\ell_{\text{peer}}$ recovers $\mathbb{1}_{\text{peer}}$ via replacing $\ell$ with $\mathbb{1}$.

**ERM with peer loss** $\quad \hat{f}_{\ell_{\text{peer}}}^* = \arg\min_{f \in \mathcal{F}} \hat{R}_{\ell_{\text{peer}}, \tilde{D}}(f) = \arg\min_{f \in \mathcal{F}} \frac{1}{N} \sum_{n=1}^{N} \ell_{\text{peer}}(f(x_n), \tilde{y}_n)$. Note again that the definition of $\ell_{\text{peer}}$ does not require the knowledge of either $e_{+1}, e_{-1}$ or $e_{+1}^*, e_{-1}^*$.

### 4.3 PROPERTY OF PEER LOSS

We now present a key property of peer loss, which shows that its risk over the noisy labels is simply an affine transformation of its true risk on clean data. We denote by $\mathbb{E}_{\mathcal{D}}[\ell_{\text{peer}}(f(X), Y)]$ the expected peer loss of $f$ when $(X, Y)$, as well as its peer samples, are drawn i.i.d. from distribution $\mathcal{D}$.

**Lemma 3.** $\mathbb{E}_{\tilde{\mathcal{D}}}[\ell_{peer}(f(X), \tilde{Y})] = (1 - e_{-1} - e_{+1}) \cdot \mathbb{E}_{\mathcal{D}}[\ell_{peer}(f(X), Y)]$.

The above Lemma states that peer loss is invariant to label noises in expectation. We have also empirically observed this effect in our experiment. Therefore minimizing it over noisy labels is equivalent to minimizing over the true distribution. The Theorems below establish the connection between $\mathbb{E}_{\mathcal{D}}[\ell_{\text{peer}}(f(X), Y)]$, the expected peer loss over clean data, with the true risk: Denote $\tilde{f}_{\mathbb{1}_{\text{peer}}}^* = \arg\min_{f \in \mathcal{F}} R_{\mathbb{1}_{\text{peer}}, \tilde{\mathcal{D}}}(f)$. With Lemma 3, we can easily prove the following:

**Theorem 2.** *[Optimality guarantee with equal prior] When* $p = 0.5$, $\tilde{f}_{\mathbb{1}_{peer}}^* \in \arg\min_{f \in \mathcal{F}} R_{\mathcal{D}}(f)$.

The above theorem states that for a class-balanced dataset with $p = 0.5$, peer loss induces the same minimizer as the one that minimizes the 0-1 loss on the clean data. Removing the constraint of $\mathcal{F}$, i.e., $\tilde{f}_{\mathbb{1}_{\text{peer}}}^* = \arg\min_f R_{\mathbb{1}_{\text{peer}}, \tilde{\mathcal{D}}}(f) \Rightarrow \tilde{f}_{\mathbb{1}_{\text{peer}}}^* = f^*$. In practice we can balance the dataset s.t. $p \to 0.5$. When $p \neq 0.5$, denote $\Delta_p = \mathbb{P}(Y = +1) - \mathbb{P}(Y = -1)$, we have the following theorem:

**Theorem 3.** *[Approximate optimality guarantee with unequal prior] When* $p \neq 0.5$, *suppose the following conditions hold: (1)* $e_{-1}, e_{+1} < 0.5$; *(2)* $(1-e) \cdot e_{-1} + e \cdot e_{+1} > e$; *(3)* $(1-e) \cdot e_{+1} + e \cdot e_{-1} > e$, *where* $e := \frac{1}{2} - \frac{\epsilon}{|\Delta_p|}$. *Then* $|R_{\mathcal{D}}(\tilde{f}_{\mathbb{1}_{peer}}^*) - \min_{f \in \mathcal{F}} R_{\mathcal{D}}(f)| \leq 2\epsilon(\bar{\ell} - \underline{\ell}), \forall \epsilon \leq |\Delta_p|/2$, *if* $\ell$ *is bounded with* $\bar{\ell}, \underline{\ell}$ *denoting its max and min.*

Condition (1) is a well-adopted assumption in the literature of learning with noisy labels. When $e_{+1}, e_{-1} > e$, we have conditions (2) and (3) hold: $(1 - e) \cdot e_{-1} + e \cdot e_{+1} > (1 - e) \cdot e + e \cdot e = e$, $(1 - e) \cdot e_{+1} + e \cdot e_{-1} > (1 - e) \cdot e + e \cdot e = e$. When $|\Delta_p|$ is small, i.e., $p$ is closer to $0.5$, this condition becomes weaker, as we will afford to have a small $\epsilon$ but also a small $e$.

**Multi-class extension** Our results in this section are largely generalizable to multi-class setting. Suppose we have $K$ classes of labels, denoting as $\{1, 2, ..., K\}$. We denote by $Q$ a transition matrix

that characterizes the relationships between noisy label $\tilde{Y}$ and the true label $Y$. The $(i, j)$ entry of $Q$ is defined as $Q_{ij} = \mathbb{P}(\tilde{Y} = j|Y = i)$. We write $Q_{ij} = q_{ij}$. For many classes of noise matrices, the $M(\cdot)$ matrix is simply a diagonal matrix. Consider the following case: suppose the noisy labels have uniform probability of flipping to a wrong class, that is, we pose the following conditions: $q_{ij} = q_{ik}$, for all $j \neq k \neq i$. This condition allows us to define $K$ new quantities $e_i = q_{ij}$ for all $i \neq j$, and $q_{ii} = 1 - \sum_{j \neq i} e_j$. We show that $M(\cdot)$ is a diagonal matrix when $\sum_{j=1}^{K} e_j < 1$, a similar condition as $e_{-1} + e_{+1} < 1$. Adapting from our proof for Lemma 3, we also have (derivation provided in Appendix) $\mathbb{E}_{\tilde{\mathcal{D}}}[\mathbb{1}_{\text{peer}}(f(X), \tilde{Y})] = (1 - \sum_{j=1}^{K} e_j) \cdot \mathbb{E}_{\mathcal{D}}[\mathbb{1}_{\text{peer}}(f(X), Y)]$. The above again will help us reach the conclusion that minimizing peer loss leads to the same minimizer on the clean data. We provide experiment results for peer loss with multi-class labels in Section 5.

## 4.4 $\alpha$-WEIGHTED PEER LOSS

We take a further look at the case with $p \neq 0.5$. Denote by $R_{+1}(f) = \mathbb{P}(f(X) = -1|y = +1)$, $R_{-1}(f) = \mathbb{P}(f(X) = +1|y = -1)$. It is easy to prove:

**Lemma 4.** *Minimizing $\mathbb{E}[\mathbb{1}_{peer}(f(X), \tilde{Y})]$ is equivalent to minimizing $R_{-1}(f) + R_{+1}(f)$.*

However, minimizing the true risk $R_{\mathcal{D}}(f)$ is equivalent to minimizing $p \cdot R_{+1}(f) + (1-p) \cdot R_{-1}(f)$, a weighted sum of $R_{+1}(f)$ and $R_{-1}(f)$. The above observation and the failure to reproduce the strong theoretical guarantee when $p \neq 0.5$ motivated us to study a $\alpha$-weighted version of peer loss, to make it robust to the case $p \neq 0.5$. We propose the following $\alpha$-*weighted peer loss* via adding a weight $\alpha \geq 0$ to the second term, the peer term:

$$(\alpha\text{-}\textbf{Peer Loss}): \quad \ell_{\alpha\text{-peer}}\big(f(x_i), \tilde{y}_i\big) = \ell(f(x_i), \tilde{y}_i) - \alpha \cdot \ell(f(x_{i_1^p}), \tilde{y}_{i_2^p}) \tag{5}$$

Denote $\mathbb{1}_{\alpha\text{-peer}}$ as $\ell_{\alpha\text{-peer}}$ when replacing $\ell$ with $\mathbb{1}$, $\tilde{f}^*_{\mathbb{1}_{\alpha\text{-peer}}} = \arg\min_{f \in \mathcal{F}} R_{\mathbb{1}_{\alpha\text{-peer}}, \tilde{\mathcal{D}}}(f)$ as the optimal classifier under $\mathbb{1}_{\alpha\text{-peer}}$, and $\Delta_{\tilde{p}} = \mathbb{P}(\tilde{Y} = +1) - \mathbb{P}(\tilde{Y} = -1)$. Then we have:

**Theorem 4.** *Let $\alpha = 1 - (1 - e_{-1} - e_{+1}) \cdot \frac{\Delta_p}{\Delta_{\tilde{p}}}$. Then $\tilde{f}^*_{\mathbb{1}_{\alpha\text{-peer}}} \in \arg\min_{f \in \mathcal{F}} R_{\mathcal{D}}(f)$.*

Denote $\alpha^* := 1 - (1 - e_{-1} - e_{+1}) \cdot \frac{\Delta_p}{\Delta_{\tilde{p}}}$. Several remarks follow: (1) When $p = 0.5$, we have $\alpha^* = 1$, we recover the earlier definition of $\ell_{\text{peer}}$. (2) When $e_{-1} = e_{+1}$, $\alpha^* = 0$, we recover $\ell$ for the clean learning setting. (3) When the signs of $\mathbb{P}(Y = 1) - \mathbb{P}(Y = -1)$ and $\mathbb{P}(\tilde{Y} = 1) - \mathbb{P}(\tilde{Y} = -1)$ are the same, $\alpha^* < 1$. Otherwise, $\alpha^* > 1$. In other words, when the noise changes the relative quantitative relationship of $\mathbb{P}(Y = 1)$ and $\mathbb{P}(Y = -1)$, $\alpha^* > 1$ and vice versa. (4) Knowing $\alpha^*$ requires certain knowledge of $e_{+1}, e_{-1}$ when $p \neq 0.5$. Though we do not claim this knowledge, this result implies tuning $\alpha^*$ (using validation data) may improve the performance.

Theorem 2 and 4 imply that performing ERM with $\mathbb{1}_{\alpha^*\text{-peer}}$: $\hat{f}^*_{\mathbb{1}_{\alpha^*\text{-peer}}} = \arg\min_f \hat{R}_{\mathbb{1}_{\alpha^*\text{-peer}}, \tilde{\mathcal{D}}}(f)$ will lead to a classifier converging to $f^*$:

**Theorem 5.** *With probability at least $1 - \delta$, $R_{\mathcal{D}}(\hat{f}^*_{\mathbb{1}_{\alpha^*\text{-peer}}}) - R^* \leq \frac{2(1+\alpha^*)}{1 - e_{-1} - e_{+1}} \sqrt{\frac{\log 2/\delta}{2N}}$.*

## 4.5 CALIBRATION AND GENERALIZATION

So far our results focused on minimizing 0-1 losses, which is hard in practice. We provide evidences of $\ell_{\text{peer}}$'s, and $\ell_{\alpha\text{-peer}}$'s in general, calibration and convexity with a generic and differentiable calibrated loss. We consider a $\ell$ that is classification calibrated, convex and $L$-Liptchitz.

**Classification calibration** describes the property that the convergence to optimality using a loss function $\ell$ would also guarantee the convergence to optimality with 0-1 loss:

**Definition 1.** *$\ell$ is classification calibrated if there $\exists$ a convex, invertible, nondecreasing transformation $\Psi_\ell$ with $\Psi_\ell(0) = 0$ s.t. $\Psi_\ell(R_{\mathcal{D}}(\tilde{f}) - R^*) \leq R_{\ell,\mathcal{D}}(\tilde{f}) - \min_f R_{\ell,\mathcal{D}}(f)$.*

Denote $f^*_\ell \in \arg\min_f R_{\ell,\mathcal{D}}(f)$. Below we provide sufficient conditions for $\ell_{\alpha\text{-peer}}$ to be calibrated.

**Theorem 6.** *$\ell_{\alpha\text{-peer}}$ is classification calibrated when either of the following two conditions holds: (1) $\alpha = 1$ (i.e., $\ell_{\alpha\text{-peer}} = \ell_{peer}$), $p = 0.5$, and $f^*_\ell$ satisfies the following: $\mathbb{E}[\ell(f^*_\ell(X), -Y)] \geq \mathbb{E}[\ell(f(X), -Y)], \forall f$. (2) $\alpha < 1, \max\{e_{+1}, e_{-1}\} < 0.5$, and $\ell''(t, y) = \ell''(t, -y)$.*

(1) states that $f_\ell^*$ not only achieves the smallest risk over $(X, Y)$ but also performs the worst on the "opposite" distribution with flipped labels $(X, -Y)$. (2) $\ell''(t, y) = \ell''(t, -y)$ is satisfied by some common loss function, such as square losses and logistic losses, as noted in (Natarajan et al., 2013),

Under the calibration condition, and denote the corresponding calibration function for $\ell_{\alpha\text{-peer}}$ as $\Psi_{\ell_{\alpha\text{-peer}}}$. Denote by $\hat{f}^*_{\ell_{\alpha\text{-peer}}} = \arg\min_{f \in \mathcal{F}} \hat{R}_{\ell_{\alpha\text{-peer}}, \tilde{D}}(f) := \frac{1}{N} \sum_{n=1}^N \ell_{\alpha\text{-peer}}(f(x_n), \tilde{y}_n)$. We have the following generalization bound:

**Theorem 7.** *The following generalization bound holds for $\ell_{\alpha^*\text{-peer}}$ with probability at least $1 - \delta$:*

$$R_\mathcal{D}(\hat{f}^*_{\ell_{\alpha^*\text{-peer}}}) - R^* \leq \frac{1}{1 - e_{-1} - e_{+1}} \cdot \Psi^{-1}_{\ell_{\alpha^*\text{-peer}}} \left( \min_{f \in \mathcal{F}} R_{\ell_{\alpha^*\text{-peer}}, \tilde{\mathcal{D}}}(f) - \min_f R_{\ell_{\alpha^*\text{-peer}}, \tilde{\mathcal{D}}}(f) \right.$$
$$\left. + 2(1 + \alpha^*) L \cdot \Re(\mathcal{F}) + 2\sqrt{\frac{\log 4/\delta}{2N}} \left( 1 + (1 + \alpha^*)(\bar{\ell} - \underline{\ell}) \right) \right),$$

*where $\Re(\mathcal{F})$ is Rademacher complexity of $\mathcal{F}$.*

**Convexity**    In experiments, we use neural networks which are more robust to non-convex loss functions. We provide sufficient conditions for $R_{\ell_{\alpha\text{-peer}}, \tilde{\mathcal{D}}}(f)$ to be convex in Appendix (Lemma 8).

## 5    EXPERIMENTS

We implemented a two-layer ReLU Multi-Layer Perceptron (MLP) for classification tasks on 10 UCI Benchmarks and applied our peer loss to update their parameters. We show the robustness of peer loss with increasing rates of label noises on 10 real-world datasets. We compare the performance of our peer loss based method with surrogate loss method (Natarajan et al., 2013) (unbiased loss correction with known error rates), symmetric loss method (Ghosh et al., 2015), DMI (Xu et al., 2019), C-SVM (Liu et al., 2003) and PAM (Khardon & Wachman, 2007), which are state-of-the-art methods for dealing with random binary-classification noises, as well as a neural network solution with binary cross entropy loss (NN). We use a cross-validation set to tune the parameters specific to the algorithms. For surrogate loss, we use the true error rates $e_{-1}$ and $e_{+1}$ instead of learning them on the validation set. Thus, surrogate loss could be considered a favored and advantaged baseline method. Accuracy of a classification algorithm is defined as the fraction of examples in the test set classified correctly with respect to the clean and true label. For given noise rates $e_{+1}$ and $e_{-1}$, labels of the training data are flipped accordingly.

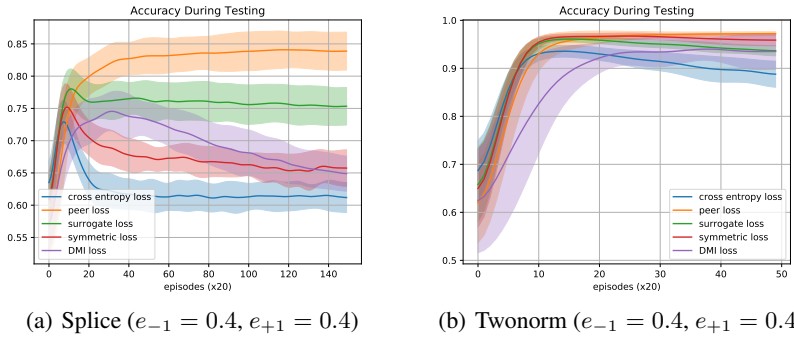

(a) Splice $(e_{-1} = 0.4, e_{+1} = 0.4)$                (b) Twonorm $(e_{-1} = 0.4, e_{+1} = 0.4)$

Figure 2: Accuracy on test set during training

A subset of the experiment results are shown in Table 1. A full table with all details can be found in Appendix. *Equalized Prior* means that we pre-sample the dataset to guarantee $p = 0.5$. For this case we used $\ell_{\text{peer}}$ without $\alpha$ (or rather $\alpha = 1$ as in $\ell_{\alpha\text{-peer}}$). For $p \neq 0.5$, we use validation dataset (using noisy labels) to tune $\alpha$. Our method is competitive across all datasets and is even able to outperform the surrogate loss method with access to the true error rates in a number of datasets, as well as symmetric loss functions (which does not require the knowledge of noise rates when error rates are symmetric) and the recently proposed information theoretical loss (Xu et al., 2019). Fig. 2 shows that our method can prevent over-fitting when facing noisy labels.

| Task | | With Prior Equalization $p = 0.5$ | | | | | Without Prior Equalization $p \neq 0.5$ | | | | |
|---|---|---|---|---|---|---|---|---|---|---|---|
| $(d, N_+, N_-)$ | $e_{-1}, e_{+1}$ | Peer | Surr | Symm | DMI | NN | Peer | Surr | Symm | DMI | NN |
| Twonorm (20,3700,3700) | 0.1, 0.3 | **0.977** | **0.968** | **0.969** | **0.974** | **0.964** | **0.977** | **0.968** | **0.969** | **0.974** | **0.964** |
| | 0.2, 0.4 | **0.976** | 0.919 | **0.959** | **0.966** | 0.911 | **0.976** | 0.919 | **0.959** | **0.966** | 0.911 |
| | 0.4, 0.4 | **0.973** | 0.934 | **0.958** | 0.936 | 0.883 | **0.973** | 0.934 | **0.958** | 0.936 | 0.883 |
| Splice (60,1527,1648) | 0.1, 0.3 | **0.919** | 0.878 | 0.851 | 0.875 | 0.811 | **0.925** | 0.885 | 0.868 | 0.889 | 0.809 |
| | 0.2, 0.4 | **0.901** | 0.832 | 0.757 | 0.801 | 0.714 | **0.912** | 0.84 | 0.782 | 0.81 | 0.725 |
| | 0.4, 0.4 | **0.819** | 0.754 | 0.657 | 0.66 | 0.626 | **0.822** | 0.755 | 0.674 | 0.647 | 0.601 |
| Heart (13,165,138) | 0.1, 0.3 | **0.833** | 0.78 | 0.777 | 0.797 | 0.756 | **0.856** | 0.802 | 0.803 | 0.83 | 0.75 |
| | 0.2, 0.4 | **0.812** | 0.768 | 0.717 | 0.788 | 0.679 | **0.856** | 0.758 | 0.725 | 0.797 | 0.693 |
| | 0.4, 0.4 | **0.75** | 0.729 | 0.654 | 0.69 | 0.595 | **0.785** | 0.728 | 0.686 | 0.711 | 0.554 |
| Diabetes (8,268,500) | 0.1, 0.3 | **0.745** | 0.707 | 0.674 | 0.72 | 0.667 | **0.778** | 0.75 | 0.738 | 0.729 | 0.727 |
| | 0.2, 0.4 | **0.755** | 0.681 | 0.634 | 0.682 | 0.596 | **0.739** | 0.705 | 0.695 | 0.707 | 0.672 |
| | 0.4, 0.4 | **0.719** | 0.645 | 0.619 | 0.637 | 0.551 | 0.651 | **0.685** | **0.68** | 0.633 | 0.583 |
| Breast (9,85,201) | 0.1, 0.3 | **0.639** | 0.563 | 0.507 | 0.529 | 0.519 | **0.727** | 0.645 | **0.709** | 0.666 | 0.648 |
| | 0.2, 0.4 | **0.63** | 0.534 | 0.482 | 0.496 | 0.538 | **0.73** | 0.674 | 0.666 | 0.58 | 0.672 |
| | 0.4, 0.4 | **0.596** | 0.519 | 0.504 | 0.526 | 0.471 | **0.677** | 0.628 | 0.545 | 0.537 | 0.529 |
| Breast (30,212,357) | 0.1, 0.3 | **0.928** | **0.922** | **0.924** | **0.934** | 0.873 | **0.956** | **0.949** | **0.943** | **0.954** | 0.92 |
| | 0.2, 0.4 | **0.93** | 0.885 | 0.844 | 0.89 | 0.844 | **0.933** | 0.898 | 0.898 | **0.918** | 0.831 |
| | 0.4, 0.4 | **0.928** | 0.867 | 0.819 | 0.746 | 0.824 | **0.908** | 0.839 | 0.817 | 0.795 | 0.673 |
| German (23,300,700) | 0.1, 0.3 | **0.701** | 0.624 | 0.614 | 0.637 | 0.581 | **0.68** | **0.693** | 0.603 | 0.605 | 0.6 |
| | 0.2, 0.4 | **0.664** | 0.59 | 0.6 | 0.618 | 0.572 | **0.676** | **0.681** | 0.537 | 0.573 | 0.535 |
| | 0.4, 0.4 | **0.606** | 0.55 | 0.573 | 0.573 | 0.556 | **0.654** | 0.632 | 0.549 | 0.611 | 0.553 |
| Waveform (21,1647,3353) | 0.1, 0.3 | **0.89** | **0.895** | **0.892** | 0.856 | 0.868 | **0.893** | **0.898** | **0.883** | 0.785 | 0.863 |
| | 0.2, 0.4 | **0.881** | **0.89** | 0.828 | 0.835 | 0.81 | **0.884** | **0.884** | 0.745 | 0.761 | 0.837 |
| | 0.4, 0.4 | **0.87** | **0.866** | **0.867** | 0.773 | 0.835 | **0.853** | **0.852** | **0.852** | 0.672 | 0.828 |
| Thyroid (5,65,150) | 0.1, 0.3 | **0.906** | **0.9** | **0.89** | 0.87 | 0.909 | **0.943** | 0.909 | 0.897 | 0.811 | **0.93** |
| | 0.2, 0.4 | **0.863** | **0.862** | **0.85** | 0.784 | 0.822 | **0.905** | 0.898 | 0.865 | 0.759 | 0.881 |
| | 0.4, 0.4 | 0.762 | 0.738 | **0.859** | 0.788 | 0.764 | 0.769 | 0.818 | **0.876** | 0.738 | 0.738 |
| Image (18,1320,990) | 0.1, 0.3 | 0.856 | 0.875 | 0.843 | **0.896** | 0.866 | 0.796 | 0.835 | **0.903** | **0.896** | 0.878 |
| | 0.2, 0.4 | 0.836 | **0.862** | 0.719 | **0.845** | 0.832 | 0.672 | 0.755 | 0.722 | **0.86** | 0.599 |
| | 0.4, 0.4 | 0.741 | 0.72 | **0.788** | 0.763 | 0.732 | **0.806** | **0.803** | **0.823** | 0.762 | 0.8 |

Table 1: Experiment results on 10 UCI Benchmarks ($N_+$, $N_-$ are the numbers of positive and negative samples). Surr: surrogate loss method (Natarajan et al., 2013); DMI: (Xu et al., 2019); Symm: symmetric loss method (Ghosh et al., 2015). Entries within 2% from the best in each row are highlighted in bold. All results are averaged across 8 random seeds. Neural-network-based methods (Peer, Surrogate, NN, Symmetric, DMI) use the same hyper-parameters. Full table with complete set of comparisons is in Appendix.

| Model | Error Rate $\epsilon = 0.2$ | Error Rate $\epsilon = 0.4$ |
|---|---|---|
| Cross Entropy | 86.67 | 82.09 |
| DMI (Xu et al., 2019) | 85.11 | 81.67 |
| Peer Loss | **87.72** | **83.81** |

Table 2: Accuracy on CIFAR-10.

**Preliminary results on multi-class classification**  We now provide some preliminary results on CIFAR-10 in Table 2. We followed the setup in (Xu et al., 2019) and used ResNet (He et al., 2016) as the underlying optimization solution. However, different from (Xu et al., 2019) whose noise only exists between specific class pairs, our noise is universal. For each class, we flip the label to any other label with a probability of $\epsilon/9$, where $\epsilon$ is the error rate and 9 is the number of other classes. We do show peer loss is competitive against Cross Entropy and DMI (Xu et al., 2019). More results and complete details are available in the Appendix.

**Conclusion**  This paper introduces peer loss, a family of loss functions that enables training a classifier over noisy labels, but without using explicit knowledge of the noise rates of labels. We provide both theoretical justifications and extensive experimental evidences.

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

ILLUSTRATION OF OUR IMPLEMENTATION OF PEER LOSS

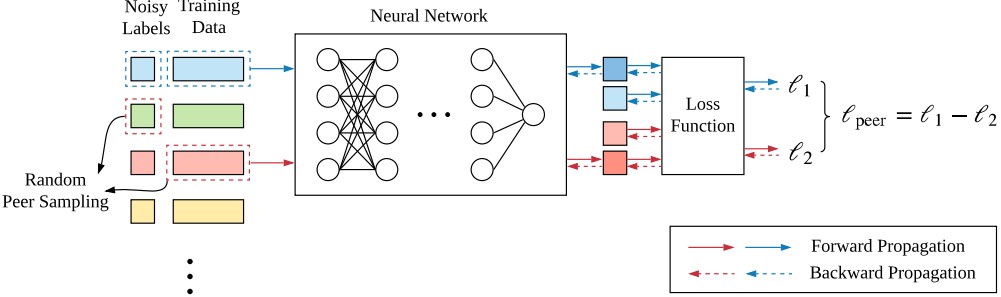

Figure 3: Illustration of peer loss method.

OTHER PEER PREDICTION FUNCTIONS

Other notable examples include quadratic and logarithmic scoring function, defined as follows:

**Example 2.** *Quadratic scoring function:*

$$S\big(r(i), r(j)\big) := 2\mathbb{P}\big(y(j) = r(j)|y(i) = r(i)\big) - \sum_{s \in \{-1, +1\}} \mathbb{P}\big(y(j) = s|y(i) = r(i)\big)^2,$$

**Example 3.** *Logarithmic scoring function:*

$$S\big(r(i), r(j)\big) := \log \mathbb{P}\big(y(j) = r(j)|y(i) = r(i)\big).$$

We know the following is true:

**Lemma 5** (Miller et al. (2005)). *S defined in Example 1 & 2 induce strict truthfulness when $y(i)$ and $y(j)$ are stochastically relevant.*

with defining stochastic relevance as follows:

**Definition 2.** $y(i)$ *and* $y(j)$ *are stochastically relevant if* $\exists\, s \in \{-1, +1\}$ *s.t.*

$$\mathbb{P}\big(y(j) = s|y(i) = +1\big) \neq \mathbb{P}\big(y(j) = s|y(i) = -1\big).$$

Similarly we conclude that when $r$ and $r^*$ are stochastic relevant, the correlated agreement scoring rule, quadratic scoring rule and logarithmic scoring rule are strictly truthful. This stochastic relevance condition essentially states that the optimal classifier is statistically different from the noisy data source $r$ on some signals. Stochastic relevance is further satisfied in the binary classification setting when $e_{-1}^* + e_{+1}^* \neq 1$, under the assumption that $e_{-1} + e_{+1} < 1$, as similarly imposed in learning with noisy labels literature (Scott et al., 2013; Natarajan et al., 2013; Scott, 2015).

**Lemma 6.** *$r$ and $r^*$ are stochastically relevant if and only if $e_{-1}^* + e_{+1}^* \neq 1$.*

*Proof.* Since $r^*$ can be written as a function of $X$ and $Y$, due to conditional independence between $r$ and $X$ (conditional on $Y$), by chain rule

$$\mathbb{P}(r^* = -1, r = +1) = \mathbb{P}(Y = +1)(1 - e_{+1})e_{+1}^* + \mathbb{P}(Y = -1)e_{-1} \cdot (1 - e_{-1}^*)$$

Since

$$\mathbb{P}(r = +1) = \mathbb{P}(Y = +1)(1 - e_{+1}) + \mathbb{P}(Y = -1) \cdot e_{-1}$$
$$\mathbb{P}(r^* = +1) = \mathbb{P}(Y = +1)(1 - e_{+1}^*) + \mathbb{P}(Y = -1) \cdot e_{-1}^*$$

We have

$$\mathbb{P}(r^* = +1, r = -1) - \mathbb{P}(r^* = +1)\mathbb{P}(r = -1)$$
$$= -\mathbb{P}(Y = +1)\mathbb{P}(Y = -1)(1 - e_{+1} - e_{-1})(1 - e_{+1}^* - e_{-1}^*) \tag{6}$$

For the binary signal case, the condition for stochastic relevance writes as follows:

$$\mathbb{P}(r = +1|r^* = +1) \neq \mathbb{P}(r = +1|r^* = -1)$$
$$\Leftrightarrow \frac{\mathbb{P}(r = +1, r^* = +1)}{\mathbb{P}(r^* = +1)} \neq \frac{\mathbb{P}(r = +1, r^* = -1)}{\mathbb{P}(r^* = -1)}$$
$$\Leftrightarrow \mathbb{P}(r = +1, r^* = +1)\mathbb{P}(r^* = -1) \neq \mathbb{P}(r = +1, r^* = -1)\mathbb{P}(r^* = +1)$$
$$\Leftrightarrow \mathbb{P}(r = +1, r^* = -1) \neq \mathbb{P}(r = +1) \cdot \mathbb{P}(r^* = -1)$$
$$\Leftrightarrow \mathbb{P}(r = +1, r^* = -1) \neq \mathbb{P}(r = +1) \cdot \mathbb{P}(r^* = -1)$$
$$\Leftrightarrow e_{-1}^* + e_{+1}^* \neq 1,$$

where the last step is a consequence of Eqn.(6).

$\square$

## PROOF FOR THEOREM 1

*Proof.* It is equivalent to prove $f^* = \text{argmax}_f \, \mathbb{E}_{(X,\tilde{Y}) \sim \tilde{\mathcal{D}}}\big[S(f(X), r)\big]$. First $S(\cdot)$ is able to elicit the Bayes optimal classifier $f^*$ ($r^*$) using $r$ implies that:

$$\mathbb{E}_{\tilde{\mathcal{D}}|Y=+1}\big[S(r^*, r)\big] > \mathbb{E}_{\tilde{\mathcal{D}}|Y=+1}\big[S(\tilde{r}^*, r)\big], \, \forall \tilde{r}^* \neq r^*$$
$$\mathbb{E}_{\tilde{\mathcal{D}}|Y=-1}\big[S(r^*, r)\big] > \mathbb{E}_{\tilde{\mathcal{D}}|Y=-1}\big[S(\tilde{r}^*, r)\big], \, \forall \tilde{r}^* \neq r^*$$

First note that the expected score of a classifier over the data distribution further writes as follows:

$$\mathbb{E}_{\tilde{\mathcal{D}}}\big[S(f(X), r)\big] = p \cdot \mathbb{E}_{\tilde{\mathcal{D}}|Y=+1}\big[S(f(X), r)\big] + (1 - p) \cdot \mathbb{E}_{\tilde{\mathcal{D}}|Y=-1}\big[S(f(X), r)\big]$$

Denote by $f'$ a sub-optimal classifier that disagrees with $f^*$ on set $\mathcal{X}_{\text{dis}}^+ = \{x|Y = +1 : f'(x) \neq f^*(x)\}$. By sub-optimality of $f'$ we know that $\epsilon := \mathbb{P}_X(X \in \mathcal{X}_{\text{dis}}^+) > 0$, as a zero measure $X_{\text{dis}}^+$ does not affect its optimality. Construct the following reporting strategy that

$$\tilde{r}^* = \begin{cases} r^*, & \text{w.p. } 1 - \epsilon \\ -r^*, & \text{w.p. } \epsilon \end{cases}$$

Not hard to check that

$$\mathbb{E}_{\tilde{\mathcal{D}}|Y=+1}\big[S(f'(X), r)\big] = \mathbb{E}_{\tilde{\mathcal{D}}|Y=+1}\big[S(\tilde{r}^*, r)\big]$$

Yet we have the following fact that

$$\mathbb{E}_{\tilde{\mathcal{D}}|Y=+1}\big[S(\tilde{r}^*, r)\big]$$
$$= (1 - \epsilon) \cdot \mathbb{E}_{\tilde{\mathcal{D}}|Y=+1}\big[S(f^*(X), r)\big]$$
$$\quad + \epsilon \cdot \mathbb{E}_{\tilde{\mathcal{D}}|Y=+1}\big[S(-f^*(X), r)\big]$$
$$< \mathbb{E}_{\tilde{\mathcal{D}}|Y=+1}\big[S(f^*(X), r)\big] \tag{7}$$

where the inequality is due to strict truthfulness of $S$ and the fact that $\epsilon > 0$. We similarly conclude that

$$\mathbb{E}_{\tilde{\mathcal{D}}|Y=-1}\big[S(\tilde{r}^*, r)\big] < \mathbb{E}_{\tilde{\mathcal{D}}|Y=-1}\big[S(f^*(X), r)\big] \tag{8}$$

Combine Eqn. (7) and (8) we conclude the proof. $\square$

## PROOF FOR LEMMA 1

*Proof.* Being categorical means
$$\mathbb{P}(r = -y|r^* = y) < \mathbb{P}(r = -y), \ y \in \{-1, +1\}$$
which further implies
$$\mathbb{P}(r = -y, r^* = y) < \mathbb{P}(r = -y)\mathbb{P}(r^* = y), \ y \in \{-1, +1\}$$
and
$$\mathbb{P}(r = y, r^* = y) > \mathbb{P}(r = y)\mathbb{P}(r^* = y), \ y \in \{-1, +1\}.$$
Consider the following fact
$$\begin{aligned}
&\mathbb{P}(r = +1, r^* = +1) \\
=&\mathbb{P}(Y = +1)\mathbb{P}(r = +1, r^* = +1|Y = +1) \\
&+ \mathbb{P}(Y = -1)\mathbb{P}(r = +1, r^* = +1|Y = -1) \\
=&\mathbb{P}(Y = +1)\mathbb{P}(r = +1|r^* = +1, Y = +1) \\
&\cdot \mathbb{P}(r^* = +1|Y = +1) \\
&+\mathbb{P}(Y = -1)\mathbb{P}(r = +1|r^* = +1, Y = -1) \\
&\cdot \mathbb{P}(r^* = +1|Y = -1)
\end{aligned}$$
Since $r^*$ can be written as a function of $X$ and $Y$, due to conditional independence between $r$ and $X$ (conditional on $Y$) we have
$$\begin{aligned}
\mathbb{P}(r = +1|r^* = +1, Y = +1) = \mathbb{P}(r = +1|Y = +1) = 1 - e_{+1}, \\
\mathbb{P}(r = +1|r^* = +1, Y = -1) = \mathbb{P}(r = +1|Y = -1) = e_{-1}
\end{aligned}$$
Therefore
$$\mathbb{P}(r = +1, r^* = +1) = \mathbb{P}(Y = +1)(1 - e_{+1})(1 - e^*_{+1}) + \mathbb{P}(Y = -1) \cdot e_{-1} \cdot e^*_{-1}$$
We also have
$$\begin{aligned}
\mathbb{P}(r = +1) = \mathbb{P}(Y = +1)(1 - e_{+1}) + \mathbb{P}(Y = -1) \cdot e_{-1} \\
\mathbb{P}(r^* = +1) = \mathbb{P}(Y = +1)(1 - e^*_{+1}) + \mathbb{P}(Y = -1) \cdot e^*_{-1}
\end{aligned}$$
Then we have
$$\begin{aligned}
&\mathbb{P}(r = +1, r^* = +1) - \mathbb{P}(r = +1)\mathbb{P}(r^* = +1) \\
=&\mathbb{P}(Y = +1)\mathbb{P}(Y = -1)(1 - e_{+1} - e_{-1})(1 - e^*_{+1} - e^*_{-1}) \\
>&0
\end{aligned}$$
when $1 > e^*_{+1} + e^*_{-1}$. $\qquad\square$

## PROOF FOR LEMMA 2

*Proof.* Again recall that
$$\mathbb{P}(r^* = +1, r = +1) = \mathbb{P}(Y = +1)(1 - e_{+1})(1 - e^*_{+1}) + \mathbb{P}(Y = -1)e_{-1} \cdot e^*_{-1}$$
$$\mathbb{P}(r = +1) = \mathbb{P}(Y = +1)(1 - e_{+1}) + \mathbb{P}(Y = -1) \cdot e_{-1}$$
$$\mathbb{P}(r^* = +1) = \mathbb{P}(Y = +1)(1 - e^*_{+1}) + \mathbb{P}(Y = -1) \cdot e^*_{-1}$$
Then we have
$$\begin{aligned}
&\mathbb{P}(r^* = +1, r = +1) - \mathbb{P}(r^* = +1)\mathbb{P}(r = +1) \\
=&\mathbb{P}(Y = +1)\mathbb{P}(Y = -1)(1 - e_{+1} - e_{-1})(1 - e^*_{+1} - e^*_{-1}) \\
>&0
\end{aligned}$$
when $1 - e_{+1} - e_{-1} > 0$, $1 - e^*_{+1} - e^*_{-1} > 0$. Interestingly this coincides with the condition imposed in (Natarajan et al., 2013). Similarly we can prove that
$$\begin{aligned}
&\mathbb{P}(r^* = +1, r = -1) - \mathbb{P}(r^* = +1)\mathbb{P}(r = -1) \\
=& - \mathbb{P}(Y = +1)\mathbb{P}(Y = -1)(1 - e_{+1} - e_{-1})(1 - e^*_{+1} - e^*_{-1}) \\
<&0
\end{aligned}$$
The other entries for $\mathbb{P}(r^* = -1, r = -1) - \mathbb{P}(r^* = -1)\mathbb{P}(r = -1)$ and $\mathbb{P}(r^* = -1, r = +1) - \mathbb{P}(r^* = -1)\mathbb{P}(r = +1)$ are symmetric. Therefore the sign matrix of above score matrix is exactly the diagonal matrix. $\qquad\square$

PROOF FOR LEMMA 3

*Proof.* We denote by $X_{i_1^p}, \tilde{Y}_{i_2^p}$ the random variable corresponding to the peer samples $x_{i_1^p}, \tilde{y}_{i_2^p}$.

First we have

$$\mathbb{E}[\ell_{\text{peer}}(f(X), \tilde{Y})] = \mathbb{E}[\ell(f(X), \tilde{Y})] - \mathbb{E}[\ell(f(X_{i_1^p}), \tilde{Y}_{i_2^p})]$$

Consider the two terms on the RHS separately.

$$\begin{aligned}
&\mathbb{E}[\ell(f(X), \tilde{Y})] \\
=&\mathbb{E}_{X,Y=-1}\big[\mathbb{P}(\tilde{Y}=-1|Y=-1)\cdot\ell(f(X),-1) + \mathbb{P}(\tilde{Y}=+1|Y=-1)\cdot\ell(f(X),+1)\big] \\
&+ \mathbb{E}_{X,Y=+1}\big[\mathbb{P}(\tilde{Y}=+1|Y=+1)\cdot\ell(f(X),+1) + \mathbb{P}(\tilde{Y}=-1|Y=+1)\cdot\ell(f(X),-1)\big] \\
=&\mathbb{E}_{X,Y=-1}\big[(1-e_{-1})\cdot\ell(f(X),-1) + e_{-1}\cdot\ell(f(X),+1)\big] \\
&+ \mathbb{E}_{X,Y=+1}\big[(1-e_{+1})\cdot\ell(f(X),+1) + e_{+1}\cdot\ell(f(X),-1)\big] \\
=&\mathbb{E}_{X,Y=-1}\big[(1-e_{-1}-e_{+1})\cdot\ell(f(X),-1) + e_{+1}\cdot\ell(f(X),-1) + e_{-1}\cdot\ell(f(X),+1)\big] \\
&+ \mathbb{E}_{X,Y=+1}\big[(1-e_{-1}-e_{+1})\cdot\ell(f(X),+1) + e_{-1}\cdot\ell(f(X),+1) + e_{+1}\cdot\ell(f(X),-1)\big] \\
=&(1-e_{-1}-e_{+1})\cdot\mathbb{E}_{X,Y}\big[\ell(f(X),y)\big] + \mathbb{E}_X\big[e_{+1}\cdot\ell(f(X),-1) + e_{-1}\cdot\ell(f(X),+1)\big]
\end{aligned}$$

And consider the second term:

$$\begin{aligned}
&\mathbb{E}[\ell(f(X_{i_1^p}), \tilde{Y}_{i_2^p})] \\
=&\mathbb{E}_X[\ell(f(X),-1)]\cdot\mathbb{P}(\tilde{Y}=-1) + \mathbb{E}_X[\ell(f(X),+1)]\cdot\mathbb{P}(\tilde{Y}=+1) \\
=&\mathbb{E}_X\big[(e_{+1}p + (1-e_{-1})(1-p))\cdot\ell(f(X),-1) + ((1-e_{+1})p + e_{-1}(1-p))\cdot\ell(f(X),+1)\big] \\
=&\mathbb{E}_X\big[(1-e_{-1}-e_{+1})(1-p)\cdot\ell(f(X),-1) + (1-e_{-1}-e_{+1})p\cdot\ell(f(X),+1)\big] \\
&+ \mathbb{E}_X\big[(e_{+1}p + e_{+1}(1-p))\cdot\ell(f(X),-1) + (e_{-1}(1-p) + e_{-1}p)\cdot\ell(f(X),+1)\big] \\
=&(1-e_{-1}-e_{+1})\cdot\mathbb{E}_X[\ell(f(X_j), \tilde{Y}_k)] + \mathbb{E}_X\big[e_{+1}\cdot\ell(f(X),-1) + e_{-1}\cdot\ell(f(X),+1)\big]
\end{aligned}$$

Thus,

$$\mathbb{E}[\ell_{\text{peer}}(f(X), \tilde{Y})] = \mathbb{E}[\ell(f(X), \tilde{Y})] - \mathbb{E}[\ell(f(X_j), \tilde{Y}_k)] = (1-e_{-1}-e_{+1})\cdot\mathbb{E}[\ell_{\text{peer}}(f(X), Y)]$$

**Multi-class extension**   Notice the following facts:

$$\mathbb{E}[\mathbb{1}(f(X), \tilde{Y})] - \mathbb{E}[\mathbb{1}(f(X_{i_1^p}), \tilde{Y}_{i_2^p})] = \mathbb{P}(\mathbb{1}(f(X) = \tilde{Y})) - \mathbb{P}(f(X_{i_1^p}) = \tilde{Y}_{i_2^p})$$

and

$$\sum_{k=1}^K \mathbb{P}(Y=k)q_{jk} = \mathbb{P}(Y=j)(1-\sum_{k\neq j}e_k) + (1-\mathbb{P}(Y=j))e_j = (1-\sum_k e_k)\mathbb{P}(Y=j) + e_j$$

$$\begin{aligned}
&\mathbb{P}(\mathbb{1}(f(X) = \tilde{Y})) \\
=&\sum_{k=1}^K \mathbb{P}(Y=k)\sum_{j=1}^K \mathbb{P}(f(X)=j|Y=k)q_{jk} \\
=&\sum_{j=1}^K\sum_{k=1}^K \mathbb{P}(f(X)=j|Y=k)\mathbb{P}(Y=k)q_{jk} \\
=&\sum_{j=1}^K \mathbb{P}(f(X)=j|Y=j)\mathbb{P}(Y=j)(1-\sum_{k\neq j}e_k) + \sum_{j=1}^K\sum_{k\neq j} \mathbb{P}(f(X)=j|Y=k)\mathbb{P}(Y=k)e_j \\
=&\sum_{j=1}^K \mathbb{P}(f(X)=j|Y=j)\mathbb{P}(Y=j)(1-\sum_{k\neq j}e_k) + \sum_{j=1}^K e_j\left(\mathbb{P}(f(X)=j) - \mathbb{P}(f(X)=j|Y=j)\mathbb{P}(Y=j)\right) \\
=&(1-\sum_k e_k)\sum_{j=1}^K \mathbb{P}(f(X)=j|Y=j)\mathbb{P}(Y=j) + \sum_{j=1}^K e_j\mathbb{P}(f(X)=j)
\end{aligned}$$

Now consider the following

$$\mathbb{P}(f(X_{i_1^p}) = \tilde{Y}_{i_2^p})$$

$$= \sum_{j=1}^{K} \mathbb{P}(f(X) = j)\mathbb{P}(\tilde{Y} = j)$$

$$= \sum_{j=1}^{K} \mathbb{P}(f(X) = j) \sum_{k=1}^{K} \mathbb{P}(Y = k)q_{jk}$$

$$= \sum_{j=1}^{K} \mathbb{P}(f(X) = j)\left((1 - \sum_{k} e_k)\mathbb{P}(Y = j) + e_j\right)$$

Therefore

$$\mathbb{E}[\mathbb{1}(f(X), \tilde{Y})] - \mathbb{E}[\mathbb{1}(f(X_{i_1^p}), \tilde{Y}_{i_2^p})]$$

$$= \mathbb{P}(\mathbb{1}(f(X) = \tilde{Y})) - \mathbb{P}(f(X_{i_1^p}) = \tilde{Y}_{i_2^p})$$

$$= (1 - \sum_{k} e_k) \sum_{j=1}^{K} (\mathbb{P}(f(X) = j|Y = j)\mathbb{P}(Y = j) - \mathbb{P}(f(X) = j)\mathbb{P}(Y = j))$$

For clean labels we have

$$\mathbb{E}[\mathbb{1}(f(X), Y)] = \sum_{j=1}^{K} \mathbb{P}(f(X) = j|Y = j)\mathbb{P}(Y = j)$$

For the second term we have

$$\mathbb{E}[\mathbb{1}(f(X_{i_1^p}), Y_{i_2^p})] = \sum_{j=1}^{K} \mathbb{P}(f(X) = j)\mathbb{P}(Y = j)$$

Therefore

$$\mathbb{E}[\mathbb{1}(f(X), Y)] - \mathbb{E}[\mathbb{1}(f(X_{i_1^p}), Y_{i_2^p})]$$

$$= \sum_{j=1}^{K} \mathbb{P}(Y = k)(\mathbb{P}(f(X) = j|Y = j)\mathbb{P}(Y = j) - \mathbb{P}(f(X) = j)\mathbb{P}(Y = j))$$

We finish the proof. □

## PROOF FOR THEOREM 2

*Proof.* From Lemma 3 we know

$$\mathbb{E}[\ell_{\text{peer}}(f(X), \tilde{Y})]$$

$$= (1 - e_{-1} - e_{+1}) \cdot \mathbb{E}[\ell_{\text{peer}}(f(X), Y)]$$

$$= (1 - e_{-1} - e_{+1}) \cdot \left(\mathbb{E}[\ell(f(X), Y)] - \mathbb{E}[\ell(f(X_{i_1^p}), Y_{i_2^p})]\right)$$

$$= (1 - e_{-1} - e_{+1}) \cdot \mathbb{E}\left[\ell(f(X), Y)] - 0.5 \cdot \mathbb{E}_X[\ell(f(X), -1)] - 0.5 \cdot \mathbb{E}_X[\ell(f(X), +1)]\right)$$

When $\ell$ is the 0-1 loss we have $\ell(f(X), -1) + \ell(f(X), +1) = 1, \forall x$, and therefore

$$\mathbb{E}[\ell_{\text{peer}}(f(X), \tilde{Y})] = (1 - e_{-1} - e_{+1}) \cdot \left(\mathbb{E}[\ell(f(X), Y)] - 1\right)$$

With above we proved $\tilde{f}^*_{\mathbb{1}_{\text{peer}}} \in \arg\min_{f \in \mathcal{F}} R_{\mathcal{D}}(f)$. □

PROOF FOR THEOREM 3

*Proof.* Our proof is inspired by our argument for $p = 0.5$. We ask the following question: if it is possible to show that $\tilde{Y}$ corresponds an error-flipped distribution of another distribution $\hat{Y}$ whose marginals $\tilde{p}_Y$ is close to or equal to 0.5. Observe the following: randomly flipping $Y$ with probability $e$ uniformly, we will have a new distribution of labels $\hat{Y}$ that satisfies:

$$\tilde{p}_Y := \mathbb{P}(\hat{Y} = +1) = \mathbb{P}(Y = +1) \cdot (1 - e) + \mathbb{P}(Y = -1) \cdot e = p(1 - 2e) + e.$$

Denote by $\epsilon$ the tolerance of $\tilde{p}_Y$: $\epsilon = |\tilde{p}_Y - 0.5|$. When $e$ sets to be: $1 - 2e = \frac{\epsilon}{|\Delta_p|}$, we have $|\tilde{p}_Y - 0.5| = \epsilon$. The next question we ask: is it possible to find parameters $\hat{e}_{-1}, \hat{e}_{+1}$:

$$\mathbb{P}(\tilde{Y} = +1|\hat{Y} = -1) = \hat{e}_{-1}, \quad \mathbb{P}(\tilde{Y} = -1|\hat{Y} = +1) = \hat{e}_{+1}$$

Note that

$$\begin{aligned}
&\mathbb{P}(\tilde{Y} = -1|Y = +1) \\
=&\mathbb{P}(\tilde{Y} = -1|\hat{Y} = +1) \cdot \mathbb{P}(\hat{Y} = +1|Y = +1) \\
&+ \mathbb{P}(\tilde{Y} = -1|\hat{Y} = -1) \cdot \mathbb{P}(\hat{Y} = -1|Y = +1) \\
=&(1 - e) \cdot \hat{e}_{+1} + e \cdot (1 - \hat{e}_{-1})
\end{aligned}$$

Similarly $\mathbb{P}(\tilde{Y} = +1|Y = -1) = (1 - e) \cdot \hat{e}_{-1} + e \cdot (1 - \hat{e}_{+1})$. Jointly we need the following equations to hold:

$$\begin{aligned}
(1 - e) \cdot \hat{e}_{+1} + e \cdot (1 - \hat{e}_{-1}) &= e_{+1} \\
(1 - e) \cdot \hat{e}_{-1} + e \cdot (1 - \hat{e}_{+1}) &= e_{-1}
\end{aligned}$$

Solving above equations we have

$$\hat{e}_{-1} = \frac{(1 - e) \cdot e_{-1} + e \cdot e_{+1}}{1 - 2e} - \frac{e}{1 - 2e}$$

For a feasible solution to $\hat{e}_{-1}, \hat{e}_{+1}$, the conditions need to satisfy that (1) $\hat{e}_{-1}, \hat{e}_{+1} \geq 0$ and (2) $\hat{e}_{-1} + \hat{e}_{+1} < 1$. First of all, from (2) we have

$$e \cdot \left(1 - (\hat{e}_{-1} + \hat{e}_{+1})\right) = e_{-1} - \hat{e}_{-1}$$

Then a necessary condition for $\hat{e}_{-1} + \hat{e}_{+1} < 1$ is

$$e_{-1} - \hat{e}_{-1} > 0 \Leftrightarrow e_{-1} < \frac{1}{2} + \frac{e_{-1}}{2(1 - 2e)}$$

This condition holds as long as $e_{-1}, e_{+1} < 0.5$. From $\hat{e}_{-1}, \hat{e}_{+1} \geq 0$ we have

$$(1 - e) \cdot e_{-1} + e \cdot e_{+1} > e, \ (1 - e) \cdot e_{+1} + e \cdot e_{-1} > e \tag{9}$$

This above jointly proves that $R_{\ell_{\alpha\text{-peer}}, \tilde{D}}(f)$ is equivalent to a peer loss defined over the noisy distribution of $\hat{y}$ with error parameters $\hat{e}_{-1}, e_{+1}$.

Denote by $f_{\mathcal{F}}^* \in \arg\min_{f \in \mathcal{F}} R_{\mathcal{D}}(f)$. From the optimality of $\tilde{f}_{\mathbb{1}_{\text{peer}}}^*$ we have

$$\begin{aligned}
&R_{\mathcal{D}}(\tilde{f}_{\mathbb{1}_{\text{peer}}}^*) - \tilde{p}_Y \cdot \mathbb{E}_X[\ell(\tilde{f}_{\mathbb{1}_{\text{peer}}}^*(X), +1)] - (1 - \tilde{p}_Y) \cdot \mathbb{E}_X[\ell(\tilde{f}_{\mathbb{1}_{\text{peer}}}^*(X), +1)] \\
&\leq R_{\mathcal{D}}(f_{\mathcal{F}}^*) - \tilde{p}_Y \cdot \mathbb{E}_X[\ell(f_{\mathcal{F}}^*(X), +1)] - (1 - \tilde{p}_Y) \cdot \mathbb{E}_X[\ell(f_{\mathcal{F}}^*(X), +1)]
\end{aligned} \tag{10}$$

Note $\forall f$:

$$\begin{aligned}
&\left|\tilde{p}_Y \cdot \mathbb{E}_X[\ell(f(X), +1)] + (1 - \tilde{p}_Y) \cdot \mathbb{E}_X[\ell(f(X), +1)]\right. \tag{11} \\
&\left.- 0.5 \cdot \mathbb{E}_X[\ell(f(X), +1)] - 0.5 \cdot \mathbb{E}_X[\ell(f(X), -1)]\right| \\
=&|\tilde{p}_Y - 0.5| \cdot \left|\mathbb{E}_X[\ell(f(X), +1)] - \mathbb{E}_X[\ell(f(X), -1)]\right| \\
\leq&\epsilon(\bar{\ell} - \underline{\ell}) \tag{12}
\end{aligned}$$

Notice that

$$
\begin{aligned}
R_{\mathcal{D}}(\tilde{f}^*_{\mathbb{1}_{\text{peer}}}) &- \tilde{p}_Y \cdot \mathbb{E}_X[\ell(\tilde{f}^*_{\mathbb{1}_{\text{peer}}}(X), +1)] - (1 - \tilde{p}_Y) \cdot \mathbb{E}_X[\ell(\tilde{f}^*_{\mathbb{1}_{\text{peer}}}(X), +1)] \\
&\leq R_{\mathcal{D}}(f^*_{\mathcal{F}}) - \tilde{p}_Y \cdot \mathbb{E}_X[\ell(f^*_{\mathcal{F}}(X), +1)] - (1 - \tilde{p}_Y) \cdot \mathbb{E}_X[\ell(f^*_{\mathcal{F}}(X), +1)] \\
&\leq R_{\mathcal{D}}(f^*_{\mathcal{F}}) - 0.5 \cdot \mathbb{E}_X[\ell(f^*_{\mathcal{F}}(X), +1)] - 0.5 \cdot \mathbb{E}_X[\ell(f^*_{\mathcal{F}}(X), +1)] + \epsilon(\bar{\ell} - \underline{\ell})
\end{aligned}
\tag{13}
$$

Combining Eqn. (10, 12, 13) we have

$$
\begin{aligned}
R_{\mathcal{D}}(\tilde{f}^*_{\mathbb{1}_{\text{peer}}}) - R_{\mathcal{D}}(f^*_{\mathcal{F}}) \leq &\tilde{p}_Y \cdot \mathbb{E}_X[\ell(f(X), +1)] + (1 - \tilde{p}_Y) \cdot \mathbb{E}_X[\ell(f(X), +1)] \\
&- 0.5 \cdot \mathbb{E}_X[\ell(f(X), +1)] - 0.5 \cdot \mathbb{E}_X[\ell(f(X), -1)] + \epsilon(\bar{\ell} - \underline{\ell}) \\
\leq &2\epsilon(\bar{\ell} - \underline{\ell})
\end{aligned}
$$

$\square$

## PROOF FOR LEMMA 4

*Proof.*

$$
\begin{aligned}
&\mathbb{E}[\mathbb{1}_{\text{peer}}(f(X), \tilde{Y})] \\
=&(1 - e_{-1} - e_{+1}) \cdot (\mathbb{P}(f(X) = -1, Y = +1) + \mathbb{P}(f(X) = +1, Y = -1) \\
&- \mathbb{P}(f(X) = -1)\mathbb{P}(Y = +1) - \mathbb{P}(f(X) = +1)\mathbb{P}(Y = -1)) \\
=&(1 - e_{-1} - e_{+1}) \cdot (pR_{+1} + (1 - p)R_{-1} \\
&- p \cdot \mathbb{P}(f(X) = 1) - (1 - p) \cdot \mathbb{P}(f(X) = -1)) \\
=&(1 - e_{-1} - e_{+1}) \cdot (pR_{+1} + (1 - p)R_{-1} \\
&- p \cdot \big(pR_{+1} + (1 - p)(1 - R_{-1})\big) - (1 - p) \cdot \big(p(1 - R_{+1}) + (1 - p)R_{-1}\big)) \\
=&2(1 - e_{-1} - e_{+1}) \cdot p(1 - p) \cdot (R_{-1} + R_{+1} - 1)
\end{aligned}
$$

$\square$

## PROOF FOR THEOREM 4

*Proof.*

$$
\begin{aligned}
&\mathbb{E}[\mathbb{1}_{\alpha\text{-peer}}(f(X), \tilde{Y})] \\
=&\mathbb{E}[\mathbb{1}(f(X), \tilde{Y})] - \alpha \cdot \mathbb{E}[\mathbb{1}(f(X_{i_1^p}), \tilde{Y}_{i_2^p})] \\
=&\mathbb{E}[\mathbb{1}_{\text{peer}}(f(X), \tilde{Y})] + (1 - \alpha) \cdot \mathbb{E}[\mathbb{1}(f(X_{i_1^p}), \tilde{Y}_{i_2^p})] - 1 \\
=&\mathbb{E}[\mathbb{1}_{\text{peer}}(f(X), \tilde{Y})] + (1 - \alpha) \cdot \big(\mathbb{P}(f(X) = -1) \cdot \mathbb{P}(\tilde{Y} = -1) + \mathbb{P}(f(X) = +1) \cdot \mathbb{P}(\tilde{Y} = +1)\big) - 1 \\
=&\mathbb{E}[\mathbb{1}_{\text{peer}}(f(X), \tilde{Y})] + (1 - \alpha) \cdot \bigg( \big(p \cdot (1 - R_{+1}) + (1 - p) \cdot R_{-1}\big) \cdot \mathbb{P}(\tilde{Y} = -1) \\
&+ \big(pR_{+1} + (1 - p)(1 - R_{-1})\big) \cdot \mathbb{P}(\tilde{Y} = +1) \bigg) - 1 \\
=&\mathbb{E}[\mathbb{1}_{\text{peer}}(f(X), \tilde{Y})] + (1 - \alpha) \cdot (\mathbb{P}(\tilde{Y} = +1) - \mathbb{P}(\tilde{Y} = -1)) \cdot (pR_{+1} - (1 - p)R_{-1}) + C \\
=&2(1 - e_{-1} - e_{+1}) \cdot p(1 - p) \cdot (R_{-1} + R_{+1} - 1) \\
&+ (1 - \alpha) \cdot (\mathbb{P}(\tilde{Y} = +1) - \mathbb{P}(\tilde{Y} = -1)) \cdot \big(pR_{+1} - (1 - p)R_{-1}\big) + C \\
=&R_{+1} \cdot \bigg( 2(1 - e_{-1} - e_{+1}) \cdot p(1 - p) + (1 - \alpha)p \cdot (\mathbb{P}(\tilde{Y} = +1) - \mathbb{P}(\tilde{Y} = -1)) \bigg) \\
&+ R_{-1} \cdot \bigg( 2(1 - e_{-1} - e_{+1}) \cdot p(1 - p) - (1 - \alpha)(1 - p) \cdot (\mathbb{P}(\tilde{Y} = +1) - \mathbb{P}(\tilde{Y} = -1)) \bigg) + C',
\end{aligned}
$$

where $C, C'$ are constants:

$$
\begin{aligned}
C &= (1 - \alpha) \cdot \Big( (1 - p) \cdot \mathbb{P}(\tilde{Y} = +1) + p \cdot \mathbb{P}(\tilde{Y} = -1) \Big) - 1 \\
C' &= C - 2(1 - e_{-1} - e_{+1}) \cdot p(1 - p)
\end{aligned}
$$

Let

$$\frac{p}{1-p} = \frac{2(1 - e_{-1} - e_{+1}) \cdot p(1-p) + (1-\alpha) \cdot p \cdot (\mathbb{P}(\tilde{Y} = +1) - \mathbb{P}(\tilde{Y} = -1))}{2(1 - e_{-1} - e_{+1}) \cdot p(1-p) - (1-\alpha) \cdot (1-p) \cdot (\mathbb{P}(\tilde{Y} = +1) - \mathbb{P}(\tilde{Y} = -1))}.$$

that

$$\alpha = 1 - (1 - e_{-1} - e_{+1}) \cdot \frac{\Delta_p}{\Delta_{\tilde{p}}}.$$

we obtain that

$$\mathbb{E}[\mathbb{1}_{\alpha\text{-peer}}(f(X), \tilde{Y})] = (1 - e_{-1} - e_{+1})\mathbb{E}[\mathbb{1}(f(X), Y)] + C', \tag{14}$$

concluding our proof. The last equation Eqn.(14) also implies the following proposition:

**Proposition 8.** *For any $f, f'$, we have*

$$\mathbb{E}_{\tilde{\mathcal{D}}}[\mathbb{1}_{\alpha\text{-peer}}(f(X), \tilde{Y})] - \mathbb{E}_{\tilde{\mathcal{D}}}[\mathbb{1}_{\alpha\text{-peer}}(f'(X), \tilde{Y})] = (1 - e_{-1} - e_{+1})\big(\mathbb{E}[\mathbb{1}(f(X), Y)] - \mathbb{E}[\mathbb{1}(f'(X), Y)]\big).$$

$\square$

## PROOF FOR THEOREM 5

*Proof.* $\forall f$, using Hoeffding's inequality with probability at least $1 - \delta$

$$|\hat{R}_{\mathbb{1}_{\alpha\text{-peer}}, \tilde{D}}(f) - R_{\mathbb{1}_{\alpha\text{-peer}}, \tilde{\mathcal{D}}}(f)|$$

$$\leq \sqrt{\frac{\log 2/\delta}{2N}}(\overline{\mathbb{1}_{\alpha-\text{peer}}} - \underline{\mathbb{1}_{\alpha-\text{peer}}})$$

$$\leq (1 + \alpha)\sqrt{\frac{\log 2/\delta}{2N}}$$

Note we also have the following:

$$R_{\mathbb{1}_{\alpha\text{-peer}}, \tilde{D}}(\hat{f}^*_{\mathbb{1}_{\alpha\text{-peer}}}) - R_{\mathbb{1}_{\alpha\text{-peer}}, \tilde{D}}(f^*_{\mathbb{1}_{\alpha\text{-peer}}})$$

$$\leq \hat{R}_{\mathbb{1}_{\alpha\text{-peer}}, \tilde{D}}(\hat{f}^*_{\mathbb{1}_{\alpha\text{-peer}}}) - \hat{R}_{\mathbb{1}_{\alpha\text{-peer}}, \tilde{D}}(f^*_{\mathbb{1}_{\alpha\text{-peer}}}) + (R_{\mathbb{1}_{\alpha\text{-peer}}, \mathcal{D}}(\hat{f}^*_{\mathbb{1}_{\alpha\text{-peer}}}) - \hat{R}_{\mathbb{1}_{\alpha\text{-peer}}, \tilde{D}}(\hat{f}^*_{\mathbb{1}_{\alpha\text{-peer}}}))$$

$$+ (\hat{R}_{\mathbb{1}_{\alpha\text{-peer}}, \tilde{D}}(f^*_{\mathbb{1}_{\alpha\text{-peer}}}) - R_{\mathbb{1}_{\alpha\text{-peer}}, \tilde{D}}(f^*_{\mathbb{1}_{\alpha\text{-peer}}}))$$

$$\leq 0 + 2 \max_f |\hat{R}_{\mathbb{1}_{\alpha\text{-peer}}, \tilde{D}}(f) - R_{\mathbb{1}_{\alpha\text{-peer}}, \tilde{\mathcal{D}}}(f)|$$

Now we show

$$R_{\mathcal{D}}(\hat{f}^*_{\mathbb{1}_{\alpha^*\text{-peer}}}) - R^*$$

$$= R_{\mathcal{D}}(\hat{f}^*_{\mathbb{1}_{\alpha^*\text{-peer}}}) - R_{\mathcal{D}}(f^*_{\mathbb{1}_{\alpha^*\text{-peer}}}) \quad \text{(Theorem 4)}$$

$$= \frac{1}{1 - e_{-1} - e_{+1}}\big(R_{\mathbb{1}_{\alpha^*\text{-peer}}, \tilde{D}}(\hat{f}^*_{\mathbb{1}_{\alpha^*\text{-peer}}}) - R_{\mathbb{1}_{\alpha^*\text{-peer}}, \tilde{D}}(f^*_{\mathbb{1}_{\alpha^*\text{-peer}}})\big) \quad \text{(Proposition 8)}$$

$$\leq \frac{2}{1 - e_{-1} - e_{+1}} \max_f |\hat{R}_{\mathbb{1}_{\alpha^*\text{-peer}}, \tilde{D}}(f) - R_{\mathbb{1}_{\alpha^*\text{-peer}}, \tilde{\mathcal{D}}}(f)|$$

$$\leq \frac{2(1 + \alpha^*)}{1 - e_{-1} - e_{+1}} \sqrt{\frac{\log 2/\delta}{2N}}.$$

We conclude the proof. $\square$

## PROOF FOR THEOREM 6

*Proof.* We start with condition (1). From Lemma 3,

$$\mathbb{E}[\ell_{\text{peer}}(f(X), \tilde{Y})] = (1 - e_{-1} - e_{+1}) \cdot \Big(\mathbb{E}[\ell(f(X), Y)] - 0.5 \cdot \mathbb{E}[\ell(f(X), -1)] - 0.5 \cdot \mathbb{E}[\ell(f(X), +1)]\Big)$$

The above further derives as

$$\mathbb{E}[\ell_{\text{peer}}(f(X), \tilde{Y})]$$

$$=(1 - e_{-1} - e_{+1}) \cdot \left( \mathbb{E}[\ell(f(X), Y)] - 0.5 \cdot \mathbb{E}[\ell(f(X), Y)] - 0.5 \cdot \mathbb{E}[\ell(f(X), -Y)] \right)$$

$$=\frac{1 - e_{-1} - e_{+1}}{2} \cdot \left( \mathbb{E}[\ell(f(X), Y)] - \mathbb{E}[\ell(f(X), -Y)] \right)$$

Denote by $c := \frac{2}{1 - e_{-1} - e_{+1}}$ we have

$$\mathbb{E}[\ell(f(X), Y)] = c \cdot \mathbb{E}[\ell_{\text{peer}}(f(X), \tilde{Y})] + \mathbb{E}[\ell(f(X), -Y)]$$

Then

$$\mathbb{E}[\ell(f(X), Y)] - \mathbb{E}[\ell(f_\ell^*(X), Y)] - (\mathbb{E}[\ell(f(X), -Y)] - \mathbb{E}[\ell(f_\ell^*(Y), -Y))]$$

$$=c \cdot (\mathbb{E}[\ell_{\text{peer}}(f(X), \tilde{Y})] - \mathbb{E}[\ell_{\text{peer}}(f_\ell^*(X), \tilde{Y})])$$

$$\leq c \cdot (\mathbb{E}[\ell_{\text{peer}}(f(X), \tilde{Y})] - \mathbb{E}[\ell_{\text{peer}}(f_{\ell_{\text{peer}}}^*(X), \tilde{Y})])$$

Further by our conditions we know

$$\mathbb{E}[\ell(f(X), Y)] - \mathbb{E}[\ell(f_\ell^*(X), Y)] - (\mathbb{E}[\ell(f(X), -Y)] - \mathbb{E}[\ell(f_\ell^*(Y), -Y))]$$
$$\geq \mathbb{E}[\ell(f(X), Y)] - \mathbb{E}[\ell(f_\ell^*(X), Y)].$$

Therefore we have proved

$$\mathbb{E}[\ell_{\text{peer}}(f(X), \tilde{Y})] - \mathbb{E}[\ell_{\text{peer}}(f_{\ell_{\text{peer}}}^*(X), \tilde{Y})] \geq \frac{1}{c} \left( \mathbb{E}[\ell(f(X), Y)] - \mathbb{E}[\ell(f_\ell^*(X), Y)] \right).$$

Since $\ell(\cdot)$ is calibrated, and according to Proposition 8 and Theorem 2:

$$\mathbb{E}_{\tilde{\mathcal{D}}}[\mathbb{1}_{\alpha\text{-peer}}(f(X), \tilde{Y})] - \mathbb{E}_{\tilde{\mathcal{D}}}[\mathbb{1}_{\alpha\text{-peer}}(f_\ell^*(X), \tilde{Y})]$$

$$=(1 - e_{-1} - e_{+1})\left( \mathbb{E}[\mathbb{1}(f(X), Y)] - \mathbb{E}[\mathbb{1}(f_\ell^*(X), Y)] \right)$$

$$\leq (1 - e_{-1} - e_{+1}) \cdot \Psi_\ell^{-1} (\mathbb{E}[\ell(f(X), Y)] - \mathbb{E}[\ell(f_\ell^*(X), Y)])$$

$$\leq (1 - e_{-1} - e_{+1}) \cdot \Psi_\ell^{-1} (c \cdot (\mathbb{E}[\ell_{\text{peer}}(f(X), \tilde{Y})] - \mathbb{E}[\ell_{\text{peer}}(f_{\ell_{\text{peer}}}^*(X), \tilde{Y})])).$$

Therefore $\Psi_{\ell_{\text{peer}}}(x) = \frac{1}{c} \Psi_\ell(\frac{x}{1 - e_{-1} - e_{+1}})$. It's straight-forward to verify that $\Psi_{\ell_{\text{peer}}}(x)$ satisfies the conditions in Definition 1. We conclude the proof.

Now we check condition (2). Again, from previously, we know the following holds for a certain $\hat{p}_y = p_y(1 - e_y) + (1 - p_y)e_{-y}$ where $p_{+1} = p, p_{-1} = 1 - p$:

$$\mathbb{E}[\ell_{\alpha\text{-peer}}(f(X), \tilde{Y})]$$

$$=\mathbb{E}[\ell(f(X), \tilde{Y}) - \alpha \cdot \ell(f(X), \tilde{Y}_k)]$$

$$=\mathbb{E}\left[ (1 - e_Y)\ell(f(X), Y) + e_Y \ell(f(X), -Y) - \alpha \cdot \hat{p}_Y \ell(f(X), Y) - \alpha \cdot (1 - \hat{p}_Y)\ell(f(X), -Y) \right]$$

$$=\mathbb{E}\left[ (1 - e_Y - \alpha\hat{p}_Y)\ell(f(X), Y) + (e_Y - \alpha \cdot (1 - \hat{p}_Y))\ell(f(X), -Y) \right]$$

Let $\phi(f(X) \cdot Y) := \ell(f(X), Y)$, we have

$$\mathbb{E}[\ell_{\alpha\text{-peer}}(f(X), \tilde{Y}))$$

$$=\mathbb{E}\left[ (1 - e_Y - \alpha\hat{p}_Y)\phi(f(X) \cdot Y) + (e_Y - \alpha \cdot (1 - \hat{p}_Y))\phi(-f(X) \cdot Y) \right]$$

$$:=\mathbb{E}[\varphi(f(X) \cdot Y)]$$

We first introduce a Theorem:

**Theorem 9** (Theorem 6, (Bartlett et al., 2006))**.** *Let $\varphi$ be convex. Then $\varphi$ is classification-calibrated if and only if it is differentiable at $0$ and $\varphi' < 0$.*

We now show that $\varphi$ is convex:

$$
\begin{aligned}
\varphi''(\beta) &= (1 - e_Y - \alpha\hat{p}_Y) \cdot \phi''(\beta) + (e_Y - \alpha \cdot (1 - \hat{p}_Y))\phi''(-\beta) \\
&= (1 - e_Y - \alpha\hat{p}_Y) \cdot \phi''(\beta) + (e_Y - \alpha \cdot (1 - \hat{p}_Y))\phi''(\beta) \\
&= (1 - e_Y - \alpha\hat{p}_Y + e_Y - \alpha \cdot (1 - \hat{p}_Y))\phi''(\beta) \\
&= (1 - \alpha)\phi''(\beta) > 0
\end{aligned}
$$

when $\alpha < 1$. The last inequality is due to the fact that $\ell$ is convex.

Secondly we show the first derivative of $\varphi$ is negative at 0: $\varphi'(0) < 0$:

$$
\begin{aligned}
\varphi'(0) &= (1 - e_Y - \alpha\hat{p}_Y) \cdot \phi'(0) - (e_Y - \alpha \cdot (1 - \hat{p}_Y))\phi'(0) \\
&= (1 - 2e_Y + \alpha(1 - 2\hat{p}_Y))\phi'(0)
\end{aligned}
\tag{15}
$$

Note that

$$
\hat{p}_y = p_y(1 - e_y) + (1 - p_y)e_{-y}
$$

Plug back to Eqn. (15) we have

$$
\begin{aligned}
\varphi'(0) &= (1 - e_Y - \alpha\hat{p}_Y) \cdot \phi'(0) - (e_Y - \alpha \cdot (1 - \hat{p}_Y))\phi'(0) \\
&= \big(1 - 2e_Y + \alpha(1 - 2\hat{p}_Y)\big)\phi'(0) \\
&= \left((1 - \alpha p_y)(1 - 2e_y) + \alpha(1 - p_y)(1 - e_{-y})\right)\phi'(0)
\end{aligned}
\tag{16}
$$

Since $(1 - \alpha p_y)(1 - 2e_y) + \alpha(1 - p_y)(1 - e_{-y}) > 0$ and $\phi'(0) < 0$ (due to calibration property of $\ell$, Theorem 6 of Bartlett et al. (2006)), we proved that $\varphi'(0) < 0$. Then based on Theorem 6 of Bartlett et al. (2006), we know $\ell_{\ell_{\alpha\text{-peer}}}$ is classification calibrated. $\qquad\square$

## PROOF FOR THEOREM 7

*Proof.* We first prove the following Rademacher complexity bound

**Lemma 7.** *Let $\Re(\mathcal{F})$ denote the Rademacher complexity of $\mathcal{F}$. $L$ denote the Lipschitz constant of $\ell$. Then with probability at least $1 - \delta$, $\max_{f \in \mathcal{F}} |\hat{R}_{\ell_{\alpha\text{-peer}}, \tilde{D}}(f) - R_{\ell_{\alpha\text{-peer}}, \tilde{D}}(f)| \leq (1 + \alpha)L \cdot \Re(\mathcal{F}) + \sqrt{\frac{\log 4/\delta}{2N}}(1 + \overline{\ell_{\alpha-peer}} - \underline{\ell_{\alpha-peer}})$.*

Note we also have the following $\forall \alpha$:

$$
\begin{aligned}
&R_{\ell_{\alpha\text{-peer}}, \tilde{D}}(\hat{f}^*_{\ell_{\alpha\text{-peer}}}) - R_{\ell_{\alpha\text{-peer}}, \tilde{D}}(f^*_{\ell_{\alpha\text{-peer}}}) \\
\leq& \hat{R}_{\ell_{\alpha\text{-peer}}, \tilde{D}}(\hat{f}^*_{\ell_{\alpha\text{-peer}}}) - \hat{R}_{\ell_{\alpha\text{-peer}}, \tilde{D}}(f^*_{\ell_{\alpha\text{-peer}}}) \\
&+ (R_{\ell_{\alpha\text{-peer}}, \tilde{D}}(\hat{f}^*_{\ell_{\alpha\text{-peer}}}) - \hat{R}_{\ell_{\alpha\text{-peer}}, \tilde{D}}(\hat{f}^*_{\ell_{\alpha\text{-peer}}})) \\
&+ (\hat{R}_{\ell_{\alpha\text{-peer}}, \tilde{D}}(f^*_{\ell_{\alpha\text{-peer}}}) - R_{\ell_{\alpha\text{-peer}}, \tilde{D}}(f^*_{\ell_{\alpha\text{-peer}}})) \\
\leq& 0 + 2\max_{f \in \mathcal{F}} |\hat{R}_{\ell_{\alpha\text{-peer}}, \tilde{D}}(f) - R_{\ell_{\alpha\text{-peer}}, \tilde{D}}(f)|
\end{aligned}
$$

Then apply the calibration condition we have

$$
\begin{aligned}
&R_{\mathcal{D}}(\hat{f}^*_{\ell_{\alpha^*\text{-peer}}}) - R^* \\
=&\frac{1}{1 - e_{-1} - e_{+1}}\big(R_{\mathbb{1}_{\alpha^*\text{-peer}},\tilde{\mathcal{D}}}(\hat{f}^*_{\ell_{\alpha^*\text{-peer}}}) - R_{\mathbb{1}_{\alpha^*\text{-peer}},\tilde{\mathcal{D}}}(f^*)\big) &\text{(Proposition 8)} \\
=&\frac{1}{1 - e_{-1} - e_{+1}}\big(R_{\mathbb{1}_{\alpha^*\text{-peer}},\tilde{\mathcal{D}}}(\hat{f}^*_{\ell_{\alpha^*\text{-peer}}}) - R_{\mathbb{1}_{\alpha^*\text{-peer}},\tilde{\mathcal{D}}}(\tilde{f}^*_{\mathbb{1}_{\alpha^*\text{-peer}}})\big) &\text{(Theorem 3)} \\
\leq&\frac{1}{1 - e_{-1} - e_{+1}}\Psi^{-1}_{\ell_{\alpha^*\text{-peer}}}\Big(\min_{f\in\mathcal{F}} R_{\ell_{\alpha^*\text{-peer}},\tilde{\mathcal{D}}}(f) - \min_{f} R_{\ell_{\alpha^*\text{-peer}},\tilde{\mathcal{D}}}(f) &\text{(Calibration of } \mathbb{1}_{\alpha^*\text{-peer}}) \\
&+ R_{\ell_{\alpha^*\text{-peer}},\tilde{\mathcal{D}}}(\hat{f}^*_{\ell_{\alpha^*\text{-peer}}}) - R_{\ell_{\alpha^*\text{-peer}},\tilde{\mathcal{D}}}(f^*_{\ell_{\alpha^*\text{-peer}}})\Big) \\
\leq&\frac{1}{1 - e_{-1} - e_{+1}}\Psi^{-1}_{\ell_{\alpha^*\text{-peer}}}\Big(\min_{f\in\mathcal{F}} R_{\ell_{\alpha^*\text{-peer}},\tilde{\mathcal{D}}}(f) - \min_{f} R_{\ell_{\alpha^*\text{-peer}},\tilde{\mathcal{D}}}(f) \\
&+ 2\max_{f\in\mathcal{F}}|\hat{R}_{\ell_{\alpha^*\text{-peer}},\tilde{D}}(f) - R_{\ell_{\alpha^*\text{-peer}},\tilde{\mathcal{D}}}(f)| \\
\leq&\frac{1}{1 - e_{-1} - e_{+1}}\Psi^{-1}_{\ell_{\alpha^*\text{-peer}}}\Big(\min_{f\in\mathcal{F}} R_{\ell_{\alpha^*\text{-peer}},\tilde{\mathcal{D}}}(f) - \min_{f} R_{\ell_{\alpha^*\text{-peer}},\tilde{\mathcal{D}}}(f) &\text{(Lemma 7)} \\
&+ 2(1+\alpha^*)L\cdot\Re(\mathcal{F}) + 2\sqrt{\frac{\log 4/\delta}{2N}}(1 + \overline{\ell_{\alpha^*\text{-peer}}} - \underline{\ell_{\alpha^*\text{-peer}}})\Big),
\end{aligned}
$$

with probability at least $1 - \delta$. $\qquad\square$

## PROOF FOR LEMMA 7

*Proof.* Due to the random sampling, via Hoeffding inequality we first have there exists some $\hat{p}_{\tilde{y}_n} \in (0,1)$, with probability at least $1 - \delta$,

$$
\begin{aligned}
\Bigg| \frac{1}{N}\sum_{n=1}^N \ell_{\alpha\text{-peer}}(f(x_n),\tilde{y}_n) &- \frac{1}{N}\sum_{n=1}^N (\ell(f(x_n),\tilde{y}_n) \\
&- \alpha\cdot\hat{p}_{\tilde{y}_n}\ell(f(x_n),\tilde{y}_n) - \alpha\cdot(1-\hat{p}_{\tilde{y}_n})\ell(f(x_n),-\tilde{y}_n))\Bigg| \\
&\leq \sqrt{\frac{\log 2/\delta}{2N}}\cdot(\overline{\ell_{\alpha-\text{peer}}} - \underline{\ell_{\alpha-\text{peer}}})
\end{aligned}
$$

Define the following loss function:

$$
\tilde{\ell}(x_n,\tilde{y}_n) := \ell(f(x_n),\tilde{y}_n) - \alpha\cdot\hat{p}_{\tilde{y}_n}\ell(f(x_n),\tilde{y}_n) - \alpha\cdot 1 - \hat{p}_{\tilde{y}_n})\ell(f(x_n),-\tilde{y}_n)
$$

Via Rademacher bound on the maximal deviation we have with probability at least $1 - \delta$

$$
\max_{f\in\mathcal{F}}\big|\hat{R}_{\tilde{\ell},\tilde{D}}(f) - R_{\tilde{\ell},\tilde{D}}(f)\big| \leq 2\cdot\Re(\tilde{\ell}\circ\mathcal{F}) + \sqrt{\frac{\log 1/\delta}{2N}} \tag{17}
$$

Since $\ell$ is $L$-Lipschitz, due to the linear combination, $\tilde{\ell}$ is $(1+\alpha)L$-Lipschitz. Based on the Lipschitz composition of Rademacher averages, we have

$$
\Re(\tilde{\ell}\circ\mathcal{F}) \leq (1+\alpha)L\cdot\Re(\mathcal{F})
$$

Therefore, via union bound, we know with probability at least $1 - 2\delta$:

$$\left| \frac{1}{N} \sum_{n=1}^{N} \ell_{\alpha\text{-peer}}(f(x_n), \tilde{y}_n) - R_{\ell_{\alpha\text{-peer}}, \tilde{D}}(f) \right|$$

$$= \left| \frac{1}{N} \sum_{n=1}^{N} \ell_{\alpha\text{-peer}}(f(x_n), \tilde{y}_n) - \hat{R}_{\tilde{\ell}, \tilde{D}}(f) + \hat{R}_{\tilde{\ell}, \tilde{D}}(f) - R_{\ell_{\alpha\text{-peer}}, \tilde{D}}(f) \right|$$

$$\leq \left| \frac{1}{N} \sum_{n=1}^{N} \ell_{\alpha\text{-peer}}(f(x_n), \tilde{y}_n) - \hat{R}_{\tilde{\ell}, \tilde{D}}(f) \right| + \left| \hat{R}_{\tilde{\ell}, \tilde{D}}(f) - R_{\ell_{\alpha\text{-peer}}, \tilde{D}}(f) \right|$$

$$\leq \sqrt{\frac{\log 2/\delta}{2N}} \cdot (\overline{\ell_{\alpha-\text{peer}}} - \underline{\ell_{\alpha-\text{peer}}}) + |\hat{R}_{\tilde{\ell}, \tilde{D}}(f) - R_{\tilde{\ell}, \tilde{D}}(f)|$$

$$\leq \sqrt{\frac{\log 2/\delta}{2N}} \cdot (\overline{\ell_{\alpha-\text{peer}}} - \underline{\ell_{\alpha-\text{peer}}}) + (1 + \alpha)L \cdot \Re(\mathcal{F}) + \sqrt{\frac{\log 1/\delta}{2N}}$$

$$\leq (1 + \alpha)L \cdot \Re(\mathcal{F}) + \sqrt{\frac{\log 2/\delta}{2N}} \cdot \left(1 + \overline{\ell_{\alpha-\text{peer}}} - \underline{\ell_{\alpha-\text{peer}}}\right)$$

In above $R_{\ell_{\alpha\text{-peer}}, \tilde{D}}(f) = R_{\varphi, \tilde{D}}(f)$ because $\ell_{\alpha\text{-peer}}$ and $\ell$ share the same expected risk by construction. Plug in the fact that $\ell_{\alpha\text{-peer}}$ is linear in $\ell$ and an easy consequence that

$$\overline{\ell_{\alpha-\text{peer}}} - \underline{\ell_{\alpha-\text{peer}}} \leq (1 + \alpha)(\bar{\ell} - \underline{\ell}),$$

let $\delta := \delta/2$, we conclude the proof.

$\square$

## PROOF FOR LEMMA 8

Nonetheless, despite the fact that $\ell_{\alpha\text{-peer}}(\cdot)$ is not convex in general, [Lemma 5, (Natarajan et al., 2013)] informs us that as long as $\hat{R}_{\ell_{\alpha\text{-peer}}, \tilde{D}}(f)$ is close to some convex function, mirror gradient type of algorithms will converge to a small neighborhood of the optimal point when performing ERM with $\ell_{\alpha\text{-peer}}$. A natural candidate for this convex function is the expectation of $\hat{R}_{\ell_{\alpha\text{-peer}}, \tilde{D}}(f)$ as $\hat{R}_{\ell_{\alpha\text{-peer}}, \tilde{D}}(f) \to R_{\ell_{\alpha\text{-peer}}, \tilde{D}}(f)$ when $N \to \infty$.

**Lemma 8.** *When $\alpha < 1, \max\{e_{+1}, e_{-1}\} < 0.5$, and $\ell''(t, y) = \ell''(t, -y)$, $R_{\ell_{\alpha\text{-peer}}, \tilde{D}}(f)$ is convex.*

*Proof.* This was proved in the proof for Theorem 6, when proving the classification calibration property of $\ell_{\alpha\text{-peer}}$ under condition (2). $\square$

## EXPERIMENT

### IMPLEMENTATION DETAILS

We implemented neural networks (LeCun et al., 2015) for classification on 10 UCI Benchmarks and applied our peer loss to update their parameters. For surrogate loss, we use the true error rates $e_{-1}$ and $e_{+1}$ instead of learning them on the validation set. Thus, surrogate loss could be considered a favored and advantaged baseline method. On each benchmark, we use the same hyper-parameters for all neural network based methods. For C-SVM, we fix one of the weights to 1, and tune the other. For PAM, we tune the margin.

### RESULTS

The full experiment results are shown in Table.**??**. *Equalized Prior* indicates that in the corresponding experiments, we resample to make sure $\mathbb{P}(Y = +1) = \mathbb{P}(Y = -1)$ and we fix $\alpha = 1$ in these experiments. Our method is competitive in all the datasets and even able to outperform the surrogate

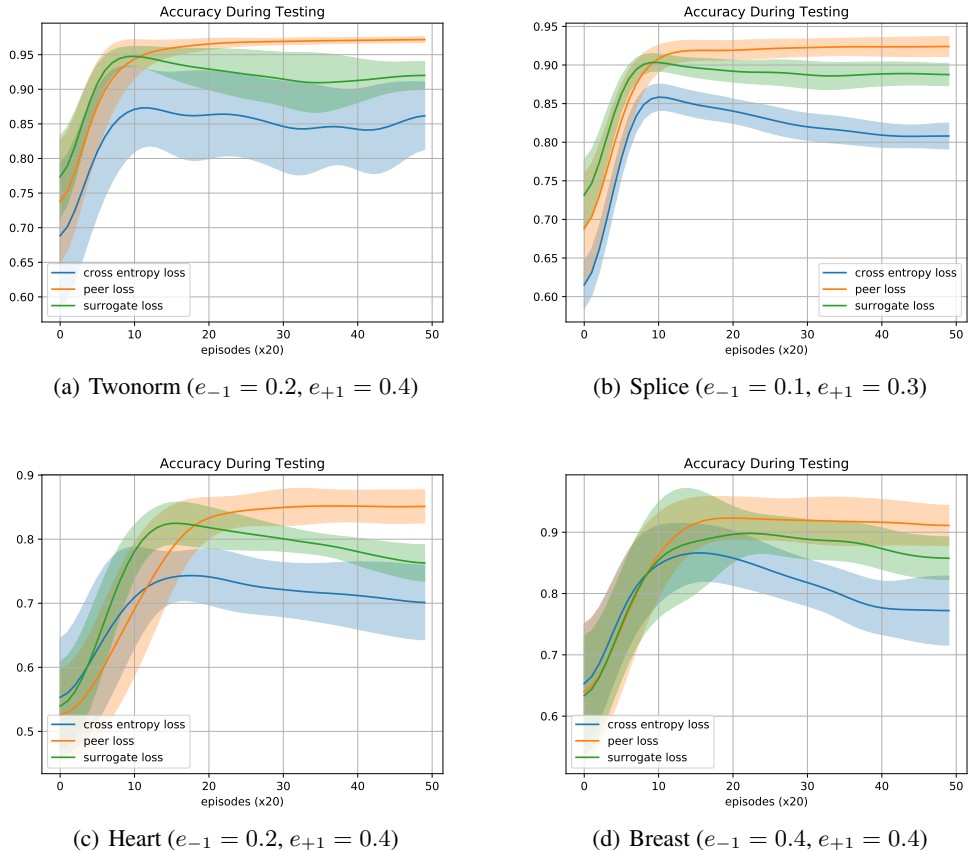

(a) Twonorm ($e_{-1} = 0.2, e_{+1} = 0.4$)

(b) Splice ($e_{-1} = 0.1, e_{+1} = 0.3$)

(c) Heart ($e_{-1} = 0.2, e_{+1} = 0.4$)

(d) Breast ($e_{-1} = 0.4, e_{+1} = 0.4$)

Figure 4: Accuracy on test set during training

loss method with access to the true error rates in most of them. C-SVM is also robust when error rates are symmetric, and is competitive in 8 datasets.

From Figure.4, we can see our peer loss can prevent over-fitting, which is also part of the reason of its achieved high robustness across different datasets and error rates.

| Task | | With Prior Equalization $p = 0.5$ | | | | | | Without Prior Equalization $p \neq 0.5$ | | | | | |
|---|---|---|---|---|---|---|---|---|---|---|---|---|---|
| $(d, N_+, N_-)$ | $e_{-1}, e_{+1}$ | Peer | Surr | Symm | DMI | NN | C-SVM | Peer | Surr | Symm | DMI | NN | C-SVM |
| Twonorm (20,3700,3700) | 0.1, 0.3 | **0.977** | **0.968** | **0.969** | **0.974** | **0.964** | **0.966** | **0.977** | **0.968** | **0.969** | **0.974** | **0.964** | **0.966** |
| | 0.2, 0.2 | **0.977** | **0.969** | **0.974** | **0.976** | **0.972** | **0.969** | **0.977** | **0.969** | **0.974** | **0.976** | **0.972** | **0.969** |
| | 0.1, 0.4 | **0.976** | **0.964** | **0.956** | **0.974** | 0.911 | 0.95 | **0.976** | **0.964** | 0.956 | **0.974** | 0.911 | 0.95 |
| | 0.2, 0.4 | **0.976** | 0.919 | **0.959** | **0.966** | 0.911 | 0.935 | **0.976** | 0.919 | **0.959** | **0.966** | 0.911 | 0.935 |
| | 0.4, 0.4 | **0.973** | 0.934 | **0.958** | 0.936 | 0.883 | 0.875 | **0.973** | 0.934 | **0.958** | 0.936 | 0.883 | 0.875 |
| Splice (60,1527,1648) | 0.1, 0.3 | **0.919** | 0.878 | 0.851 | 0.875 | 0.811 | **0.928** | **0.925** | 0.885 | 0.868 | 0.889 | 0.809 | **0.933** |
| | 0.2, 0.2 | **0.918** | 0.874 | 0.879 | 0.888 | 0.819 | **0.931** | **0.927** | 0.876 | 0.906 | 0.885 | 0.812 | **0.941** |
| | 0.1, 0.4 | **0.914** | 0.86 | 0.757 | 0.842 | 0.743 | 0.891 | **0.925** | 0.862 | 0.777 | 0.852 | 0.754 | 0.898 |
| | 0.2, 0.4 | **0.901** | 0.832 | 0.757 | 0.801 | 0.714 | 0.807 | **0.912** | 0.84 | 0.782 | 0.81 | 0.725 | 0.824 |
| | 0.4, 0.4 | **0.819** | 0.754 | 0.657 | 0.66 | 0.626 | 0.767 | **0.822** | 0.755 | 0.674 | 0.647 | 0.601 | 0.76 |
| Heart (13,165,138) | 0.1, 0.3 | **0.833** | 0.78 | 0.777 | 0.797 | 0.756 | 0.753 | **0.856** | 0.802 | 0.803 | 0.83 | 0.75 | 0.788 |
| | 0.2, 0.2 | **0.821** | 0.762 | 0.795 | **0.801** | 0.75 | 0.717 | **0.856** | 0.813 | 0.793 | 0.826 | 0.769 | 0.796 |
| | 0.1, 0.4 | **0.827** | 0.777 | 0.714 | 0.779 | 0.717 | 0.744 | **0.859** | 0.815 | 0.725 | 0.814 | 0.723 | 0.677 |
| | 0.2, 0.4 | 0.812 | 0.768 | 0.717 | 0.788 | 0.679 | 0.714 | **0.856** | 0.758 | 0.725 | 0.797 | 0.693 | 0.704 |
| | 0.4, 0.4 | **0.75** | 0.729 | 0.654 | 0.69 | 0.595 | 0.688 | **0.785** | 0.728 | 0.686 | 0.711 | 0.554 | 0.698 |
| Diabetes (8,268,500) | 0.1, 0.3 | **0.745** | 0.707 | 0.674 | 0.72 | 0.667 | 0.67 | **0.778** | 0.75 | 0.738 | 0.729 | 0.727 | 0.726 |
| | 0.2, 0.2 | **0.755** | 0.708 | 0.72 | 0.729 | 0.671 | **0.745** | 0.759 | 0.736 | 0.753 | **0.743** | 0.706 | **0.759** |
| | 0.1, 0.4 | **0.745** | 0.682 | 0.612 | 0.701 | 0.627 | 0.568 | **0.777** | 0.724 | 0.694 | 0.713 | 0.71 | 0.688 |
| | 0.2, 0.4 | **0.755** | 0.681 | 0.634 | 0.682 | 0.596 | 0.59 | **0.739** | 0.705 | 0.695 | 0.707 | 0.672 | 0.7 |
| | 0.4, 0.4 | **0.719** | 0.645 | 0.619 | 0.637 | 0.551 | 0.654 | 0.651 | **0.685** | 0.68 | 0.633 | 0.583 | **0.702** |
| Breast (9,85,201) | 0.1, 0.3 | **0.639** | 0.563 | 0.507 | 0.529 | 0.519 | 0.529 | **0.727** | 0.645 | **0.709** | 0.666 | 0.648 | 0.698 |
| | 0.2, 0.2 | **0.659** | 0.606 | 0.537 | 0.548 | 0.534 | 0.615 | **0.698** | 0.661 | 0.655 | 0.627 | 0.623 | **0.695** |
| | 0.1, 0.4 | **0.587** | **0.577** | 0.504 | 0.504 | 0.519 | 0.553 | **0.735** | 0.654 | 0.685 | 0.621 | 0.66 | 0.698 |
| | 0.2, 0.4 | **0.63** | 0.534 | 0.482 | 0.496 | 0.538 | 0.538 | **0.73** | 0.674 | 0.666 | 0.58 | 0.672 | 0.698 |
| | 0.4, 0.4 | **0.596** | 0.519 | 0.504 | 0.526 | 0.471 | 0.51 | 0.677 | 0.628 | 0.545 | 0.537 | 0.529 | **0.698** |
| Breast (30,212,357) | 0.1, 0.3 | **0.928** | 0.922 | 0.924 | **0.934** | 0.873 | **0.924** | **0.956** | 0.949 | 0.943 | **0.954** | 0.92 | **0.943** |
| | 0.1, 0.4 | **0.932** | **0.938** | **0.937** | **0.944** | 0.83 | 0.85 | **0.951** | 0.929 | **0.946** | **0.941** | 0.898 | 0.929 |
| | 0.2, 0.2 | 0.928 | 0.904 | 0.835 | 0.897 | 0.887 | **0.961** | 0.952 | 0.952 | 0.897 | **0.942** | **0.955** | 0.946 |
| | 0.2, 0.4 | **0.93** | 0.885 | 0.844 | 0.89 | 0.844 | 0.865 | **0.933** | 0.898 | 0.898 | **0.918** | 0.831 | 0.862 |
| | 0.4, 0.4 | **0.928** | 0.867 | 0.819 | 0.746 | 0.824 | 0.855 | **0.908** | 0.839 | 0.817 | 0.795 | 0.673 | 0.866 |
| German (23,300,700) | 0.1, 0.3 | **0.701** | 0.624 | 0.614 | 0.637 | 0.581 | 0.611 | **0.68** | **0.693** | 0.603 | 0.605 | 0.6 | 0.671 |
| | 0.2, 0.2 | **0.689** | 0.65 | 0.647 | 0.623 | 0.611 | 0.664 | 0.702 | 0.693 | 0.704 | 0.62 | 0.6 | **0.738** |
| | 0.1, 0.4 | **0.696** | 0.642 | 0.587 | 0.63 | 0.562 | 0.55 | 0.667 | **0.693** | 0.54 | 0.594 | 0.54 | 0.553 |
| | 0.2, 0.4 | **0.664** | 0.59 | 0.6 | 0.618 | 0.572 | 0.469 | 0.676 | **0.681** | 0.537 | 0.573 | 0.535 | 0.581 |
| | 0.4, 0.4 | **0.606** | 0.55 | 0.573 | 0.573 | 0.556 | 0.572 | 0.654 | 0.632 | 0.549 | 0.611 | 0.553 | **0.696** |
| Waveform (21,1647,3353) | 0.1, 0.3 | 0.89 | 0.895 | 0.892 | 0.856 | 0.868 | 0.862 | **0.893** | **0.898** | 0.883 | 0.785 | 0.863 | **0.878** |
| | 0.2, 0.2 | 0.883 | 0.899 | **0.9** | 0.861 | **0.894** | **0.886** | **0.901** | **0.899** | **0.894** | 0.792 | **0.898** | **0.897** |
| | 0.1, 0.4 | **0.884** | **0.893** | 0.762 | 0.856 | 0.771 | 0.804 | **0.888** | **0.894** | 0.703 | 0.778 | 0.821 | 0.821 |
| | 0.2, 0.4 | **0.881** | **0.89** | 0.828 | 0.835 | 0.81 | 0.795 | **0.884** | **0.884** | 0.745 | 0.761 | 0.837 | 0.837 |
| | 0.4, 0.4 | **0.87** | **0.866** | **0.867** | 0.773 | 0.835 | 0.776 | **0.853** | **0.852** | **0.852** | 0.672 | 0.828 | **0.848** |
| Thyroid (5,65,150) | 0.1, 0.3 | 0.906 | **0.9** | 0.89 | 0.87 | **0.909** | 0.881 | **0.943** | 0.909 | 0.897 | 0.811 | **0.93** | **0.924** |
| | 0.2, 0.2 | **0.913** | 0.894 | **0.907** | 0.897 | 0.899 | **0.918** | 0.905 | 0.905 | 0.905 | 0.91 | **0.936** | **0.936** |
| | 0.1, 0.4 | **0.875** | 0.862 | 0.834 | 0.784 | **0.88** | 0.869 | 0.902 | **0.924** | 0.856 | 0.75 | **0.919** | **0.917** |
| | 0.2, 0.4 | **0.863** | **0.862** | **0.85** | 0.784 | 0.822 | 0.781 | **0.905** | 0.898 | 0.865 | 0.759 | 0.881 | **0.92** |
| | 0.4, 0.4 | 0.762 | 0.738 | **0.859** | 0.788 | 0.764 | 0.781 | 0.769 | 0.818 | **0.876** | 0.738 | 0.738 | 0.837 |
| Image (18,1320,990) | 0.1, 0.3 | 0.856 | 0.875 | 0.843 | **0.896** | 0.866 | **0.892** | 0.796 | 0.835 | **0.903** | 0.896 | 0.878 | **0.892** |
| | 0.2, 0.2 | **0.9** | 0.835 | **0.911** | 0.894 | **0.908** | **0.912** | **0.931** | 0.896 | 0.917 | 0.883 | **0.934** | 0.908 |
| | 0.1, 0.4 | 0.723 | 0.841 | 0.705 | **0.881** | 0.799 | 0.785 | 0.717 | 0.806 | 0.679 | **0.888** | 0.825 | 0.808 |
| | 0.2, 0.4 | 0.836 | **0.862** | 0.719 | **0.845** | 0.832 | 0.802 | 0.672 | 0.755 | 0.722 | **0.86** | 0.599 | 0.825 |
| | 0.4, 0.4 | 0.741 | 0.72 | 0.788 | 0.763 | 0.732 | **0.834** | 0.806 | 0.803 | 0.823 | 0.762 | 0.8 | **0.86** |

Table 3: Experiment Results on 10 UCI Benchmarks. Entries within 2% from the best in each row are in bold. Surr: surrogate loss method (Natarajan et al., 2013); DMI: (Xu et al., 2019); Symm: symmetric loss method (Ghosh et al., 2015). All method-specific parameters are estimated through cross-validation. The pro- posed method (Peer) are competitive across all the datasets. Neural-network-based methods (Peer, Surrogate, NN, Symmetric, DMI) use the same hyper-parameters. All the results are averaged across 8 random seeds.

