# OpenReview forum: "Peer Loss Functions: Learning from Noisy Labels without Knowing Noise Rates"
_ICLR.cc/2020/Conference — Reject_

### Official Review · AnonReviewer1 · 2019-10-17
**Official Blind Review #1**

**Rating:** 3

**Review:**

This paper proposed peer loss function for learning with noisy labels, combining two areas learning with noisy labels and peer prediction together. The novelty and the significance are both borderline (or below). There are 4 major issues I have found so far.

References: Looking at section 1.1 the related work, the references are a bit too old. While I am not sure about the area of peer prediction, in the area of learning with noisy labels (in a general sense), there were often 10 to 15 papers from every NeurIPS, ICML, ICLR and CVPR in recent years. The authors didn't survey the literature after 2016 at all... Nowadays most papers focus on sample selection/reweighting and label correction rather than loss correction in this area, but there are still many recent papers on designing more robust losses, see https://arxiv.org/abs/1805.07836 (NeurIPS 2018 spotlight), https://openreview.net/forum?id=rklB76EKPr and references therein. Note also that some label-noise related papers may not have the term label noise or noisy labels in the title, for example, https://openreview.net/forum?id=B1xWcj0qYm (ICLR 2019).

Motivation: The motivating claim "existing approaches require practitioners to specify noise rates" is wrong... Many loss correction methods can estimate the transition matrix T (which is indispensable in any loss correction) without knowing the noise rate, when there are anchor points or even no anchor points in the noisy training data. See https://arxiv.org/abs/1906.00189 (NeurIPS 2019) and references therein. See also the public comment posted by Nontawat when a special symmetric condition is assumed on the surrogate loss function.

Novelty: The paper introduced peer prediction, an area in computational economics and algorithmic game theory, to learning with noisy labels. This should be novel (to the best of my knowledge) and I like it! However, the obtained loss is very similar to the general loss correction approach, see https://arxiv.org/abs/1609.03683 (CVPR 2017 oral). This fact undermines the novelty of the paper, significantly. The authors should clarity the connection to and the difference from the loss correction approach.

Significance: The proposed method focuses on binary classification, otherwise the paper will be much more significant! Note that the backward and forward corrections can both be applied to multi-class classification. Moreover, similar to many theory papers, the experiments are too simple, where single-hidden-layer neural networks were trained on 10 UCI benchmark datasets. I have to say this may not be enough for ICLR that should be a more deep learning conference.

**Experience Assessment:**

I have published in this field for several years.

**Review Assessment: Checking Correctness Of Derivations And Theory:**

I assessed the sensibility of the derivations and theory.

**Review Assessment: Checking Correctness Of Experiments:**

I assessed the sensibility of the experiments.

**Review Assessment: Thoroughness In Paper Reading:**

I read the paper thoroughly.

---

> ### Author Response · Authors · 2019-11-06
> **clarification of our motivation, novelty and significance; peer loss operates without estimating transition matrices ; added experiment on CIFAR-10**
>
> We thank the reviewer for detailed comments, and for providing pointers to the more recent literature. We want to clarify our motivation, novelty and significance. We do think the reviewer might have missed the main novelty and contribution of peer loss in that peer loss operates without the need of estimating the transition matrix T.
>
> Reference:
>
> We apologize for not surveying sufficiently the most recent results. We wanted to focus on comparing with the more classical results. We are working on an updated related work section based on the reviewer’s and Nontawat’s comments. We thank the reviewer for relevant pointers.
>
>
> Motivation:
>
> When we claim “existing approaches require practitioners to specify noise rates”, it includes the cases when the transition matrix T need to and can be estimated. Probably our claim was not clear - we wanted to say that these existing methods require *explicit* knowledge of T, so in practice, the practitioners will need to estimate these Ts and plug in. We are indeed aware of these line of work. But this was exactly one of our main motivations! We felt this requirement of additional estimation steps might complicate the learning process, and the additional errors introduced via learning these parameters are concerning too.
>
> Peer loss does *not* require the knowledge or estimation of these Ts, and operates without the need of *specifying* the noise rates. We now start to think that a better way to position our paper is to say we contribute to “learning with noisy labels without specifying the error rates”.
>
> We do not claim that our results will challenge all existing works that require estimation of the transition matrices, we are simply trying to provide an alternative that operates without these estimates and when these estimates might not be reliable/available. Nonetheless, in our experiments, we provide evidences that even we give the surrogate loss function method (e.g., [Natarajan et al 13]) a perfect estimate of the noise rates, peer loss has shown advantages.
>
> We are glad the reviewer 2 in fact mentioned the challenge of estimating instance dependent transition matrix - part of the reason that we are excited about peer loss is that peer loss removes the need of estimating transition matrix and it opens up a new possibility in handling instance-dependent label noises.
>
> We acknowledge the comments, from both Nontawat and the reviewer, that for symmetric error rates case  there exist methods that do not require the knowledge of error rates. We focus on asymmetric error rate cases. We are updating our draft and will clarify this.
>
>
> Novelty:
>
> Thank you for pointing out the CVPR 2017 paper (we now cited). There seems to be a misunderstanding - unless we are mistaking, the proposed loss function and the correction procedure therein would require the transition matrix T to perform backward and forward corrections. In fact, the proposed loss correction method is derived from the surrogate loss function literature (again, e.g., [Natarajan et al 13]), which is one of the baseline method we compare to. We want to emphasize again our operation of peer loss does *not* require this knowledge of T. We do not see any further connection, but would love to hear a more specific pointer if the reviewer disagrees. Please let us know, thanks.
>
> Our connection of using peer prediction in this learning with noisy label setting frees up the requirement of estimating the transition matrix, which we believe is novel (this was acknowledged in other reviewer’s comments, including R3 and Nontawat’s).
>
> *Updated comments*: After reading the CVPR more carefully, the paper did mention the Hessian of ReLU is invariant to noises, so minimizing it will not need the specification of noise rates. First, this is not the same as the proposed loss correction approach, which is the focus of the paper; Second, it is acknowledged in the paper that " this does not provide any assurance on minima: indeed, stationary points may change location due to label noise."
>
>
> Significance:
>
> Again because of that peer loss does not require the knowledge of transition matrix, our method generalizes to multi-class labels easily. In an earlier draft we do provide a justification, but we wanted to stay focused with our binary setting, since our results are already dense. In the updated version, we will add the preliminary results back.
>
> *Update*: we have restructured our presentation, and have added an experiment on CIFAR-10.
>
> We hope the above helps clarify our contributions. Thank you for reading our paper, and we are happy to discuss further.

---

### Official Review · AnonReviewer2 · 2019-10-23
**Official Blind Review #2**

**Rating:** 3

**Review:**

The paper studies the label noise problem with the motivation of without estimating the flip rate or transition matrix. This is an interesting direction for dealing with label noise. Most of the previous studies need either estimate the transition matrix or put restrictions on it, e.g., to be symmetric. A very related work to this paper: L_{DMI}: A Novel Information-theoretic Loss Function for Training Deep Nets Robust to Label Noise, where no restrictions have been made on the class-dependent transition matrix and the proposed method does not need to estimate the transition matrix. The authors may need to discuss the paper.

The paper is not well-presented. I tried several times to go through the details but failed. The reasons are, e.g., (1) notation is not clear, e.g., R(X). \tilde{Y}, and R have been abused. (2) Intuitive explanations are limited. (3) Lots of details can be put on the appendix, keeping the main part to have a strong and clear logic.

It seems the theories of the paper depends on a very strong assumption, i.e., "Suppose S(.) is able to elicit the Bayes optimal classifier f^*".  By quickly go through the proofs in the appendix, it seems there is a strong connection to the paper L_{DMI}.

Some claims are strong. In the literature, with a mild assumption, the class-dependent transition matrix can accurately be estimated just from noisy data with theoretical guarantees. There are also methods proposed for this. Estimating class-dependent transition matrix is not a bottleneck. I think the challenge is about how to learn instance-dependent transition matrix.

Overall, this is an interesting paper but needs to be improved.

**Experience Assessment:**

I have published in this field for several years.

**Review Assessment: Checking Correctness Of Derivations And Theory:**

I assessed the sensibility of the derivations and theory.

**Review Assessment: Checking Correctness Of Experiments:**

I assessed the sensibility of the experiments.

**Review Assessment: Thoroughness In Paper Reading:**

I read the paper at least twice and used my best judgement in assessing the paper.

---

> ### Author Response · Authors · 2019-11-06
> **clarification of our contributions, assumptions and claims**
>
> We thank the reviewer for his/her comments. We will clarify our contribution in comparing to L_{DMI}, our assumptions made for the theoretical results, and our claims.
>
>
> Compare to L_{DMI}:
>
> While we are working hard to compare us with L_{DMI} and update our draft, we’d like to mention that the pointed article was submitted to arXiv on Sep 8, 2019, which was roughly two weeks before ICLR deadline. We feel that would be too fresh for us to include a thorough comparison - around the point, we have pretty much finalized our results and were focusing on writing up our draft.
>
> We have received a public comment on L_{DMI}, and have commented publicly about the differences. We copy the responses at the end too. Since received the comment, we have looked into the codes shared by the authors of L_{DMI}. We do realize the experiments presented therein largely focused on simple noise model, with noises being either added to only one class (one single noise parameter, and this makes the problem much easier! as one class has entirely clean labels, and the other class still has dominantly clean label!), and symmetric noises (different class labels have the same error rates, again one noise parameter). We do think our results provide evidence that peer loss is robust to different asymmetric noise settings.
>
>
> Our assumptions:
>
> Inly our conceptual results in Section 3 require S to elicit Bayes optimal classifier. Our main results on the specific forms of peer losses in Section 4 do *not* require any of such. Our theoretical results and properties hold for any hypothesis class.
>
> For instance, in Theorem 3,4 (Sec 4.1), Theorem 5 (Sec 4.2), the results are over a generic hypothesis class $\mathcal F$.
>
>
> On our claim:
>
> Probably our claim was not clear - we wanted to say that these existing methods require *explicit* knowledge of T, so in practice, the practitioners will need to estimate these Ts and plug in. We are indeed aware of these line of work. But this was exactly one of our main motivations! We felt this requirement of additional estimation steps might complicate the learning process, and the additional errors introduced via learning these parameters are concerning too.
>
> Peer loss does *not* require the knowledge or estimation of these Ts, and operates without the need of *specifying* the noise rates. We now start to think that a better way to position our paper is to say we contribute to “learning with noisy labels without specifying the error rates”.
>
> We do not claim that our results will challenge all existing works that require estimation of the transition matrices, we are simply trying to provide an alternative that operates without these estimates and when these estimates might not be reliable/available. Nonetheless, in our experiments, we provide evidence that peer loss has shown advantages, even we give the surrogate loss function method (e.g., [Natarajan et al 13]) a perfect estimate of the noise rates.
>
> We are glad the reviewer mentioned the challenge of estimating instance dependent transition matrix - part of the reason that we are excited about peer loss is that peer loss removes the need of estimating transition matrix and it opens up a new possibility in handling instance-dependent label noises.
>
> We hope the above clarifies our contribution. Happy to discuss further.
>
> —————---------------------------------
> Earlier public comments to the difference between our work and L_{DMI}:
>
> - We aimed for a simple-to-optimize loss function that can easily adapt to existing ERM solutions. After the submission, we have observed other successes in adapting peer loss to more sophisticated neural network solutions, differentially private ERM, and semi-supervised learning etc. [1] seems to require estimations of a joint distribution matrix, and then to invoke computing a certain information theoretical measure.
>
> - With above concern, it is not entirely clear to us about the sample complexity requirement in [1], and the sensitivity to noises in this estimation. We do provide calibration guarantees and generalization bounds.
>
> - We provide conditions when the loss functions are convex. In general, we do think computationally peer loss functions are easy to optimize with, in comparing to information theoretical measures.
>
> - We provide extensive comparisons with the state-of-the-art learning with noisy label approaches (which even have access to the error rate information).
>
> - We provide theoretical guarantees for both Bayes' optimal classifiers and general hypothesis classes, and establish a broad connection between learning with noisy labels with peer prediction score functions. It is true we have focused on CA, but the connection implies the possibility of applying other peer prediction functions.

---

> > ### Author Response · Authors · 2019-11-06
> > **R was indeed a bit abused in Section 3; thank you!**
> >
> > Thank for pointing out that R was a bit abused in Section 3. We will find another notation for it in the updated draft.

---

> > ### Comment · AnonReviewer2 · 2019-11-14
> > **The organization and presentation could be further improved.**
> >
> > Thanks for the detailed reply! I do agree with most of them. I still have the concern that the readability of the paper could be further improved. The presentation of the technique and theory parts are with minor changes in the updated version. "Probably your claims are clear". However,  an easy-to-understand writing is much appreciated from the readers' perspective.
> >
> > After carefully reading the paper L_{DMI}, I found that we may not avoid estimating the transition matrix without any price. Given the noisy class posteriors, we need to carefully balance the tradeoff between the transition matrix and the clean class posteriors. In other words, how can you make sure that the clean class posterior is identifiable? This point should be clearly presented and discussed for the thread of methods avoiding estimating the transition matrices.

---

> > > ### Author Response · Authors · 2019-11-14
> > > **working on that! just uploaded a new version, but will continue updating it**
> > >
> > > Great! We are glad that we were able to clarify some concerns.
> > >
> > > Yes indeed we have been making some structural changes, and adding explanations. We suffered from page limit and are working on the best way to deliver the explanations. Hopefully we will finish before the deadline!
> > >
> > > In the current revision (uploaded now),
> > >
> > > - we added a paragraph (top of page 2) to briefly introduce why peer loss can get rid of estimation of noise rates.
> > > - at the end of Sec 3 (page 5), we added a paragraph to explain the connection between Sec 3 and 4.
> > > - we restructured Sec 4: now in 4.1, we present some preparation and explain the effects of peer term. 4.2 focuses on peer loss only. We added an example and explanation of why we do not need the explicit knowledge of noises rates at the top of page 7.
> > > - Sec 4.3 now has an interpretation of Lemma 3 and how it helps.
> > >
> > > We think probably Sec 4.1 - 4.3 are the most important ones to understand the basic ideas, while 4.4 and 4.5 focus on the technical properties.
> > >
> > > We are checking with the reviewer to see whether we are on the right direction about this and would appreciate further comments/suggestions (before the response deadline) on improving readability (but we understand if not, given the short time left).

---

### Official Review · AnonReviewer3 · 2019-10-23
**Official Blind Review #3**

**Rating:** 8

**Review:**

This paper studies the problem of learning classifiers from noisy data without specifying the noise rates. Inspired by the literature of peer prediction, the authors propose peer loss. First, a scoring function is introduced, minimizing which we can elicit the Bayes optimal classifier f*. Then the authors use the setting of CA to induces a scoring matrix, and then the peer loss. Moreover, this paper explores the theoretical properties of peer loss when p=0.5. In particular, the authors propose \alpha weighted peer loss to provide strong theoretical guarantees of the proposed ERM framework. The calibration and generalization abilities are also discussed in section 4.3. Finally, empirical studies show that the propose peer loss indeed remedies the difficulty of determining the noise rates in noisy label learning.

This paper is well written. The theoretical properties of the proposed peer loss are thoroughly explored. The motivation is rational with a good theoretical guarantee, i.e. Theorem 1. Moreover, the tackled problem, i.e. avoiding specifying the noise rates, is significant to the community.

Nevertheless, Some parts of this paper may be confusing:
- The computation of the scoring matrix delta is not that clear. Can the authors provide the detailed computation steps of the example?
- In the proof of Lemma 6, can the authors provide a proof sketch of the equivalence of the last two equations?
- Third, where is the definition of p?

In the experiments, the authors propose to tuning the hyperparameter alpha. I would be appreciated if the authors provide the sensitivity experiments of alpha to show its fluence for the final prediction.

Though I'm not that familiar with learning from noisy labels, I think it is a good paper and I suggest to accept.

**Experience Assessment:**

I do not know much about this area.

**Review Assessment: Checking Correctness Of Derivations And Theory:**

I carefully checked the derivations and theory.

**Review Assessment: Checking Correctness Of Experiments:**

I carefully checked the experiments.

**Review Assessment: Thoroughness In Paper Reading:**

I read the paper thoroughly.

---

> ### Author Response · Authors · 2019-11-06
> **clarifications & thank you for acknowledging our contributions**
>
> We thank the reviewer for the comments, and for acknowledging our contribution of “avoiding specifying the noise rates, is significant to the community.” Below we clarify
>
> Detailed examples for computing Delta:
>
> First of all, we compute the marginals of $f^*$ and $\tilde{Y}$:
> $P \bigl(f^*(x)=-1\bigr) = P \bigl(f^*(x)=-1|Y=-1\bigr) P(Y=-1) + P \bigl(f^*(x)=-1|Y=+1\bigr) P(Y=+1) = (1-e^*_{-1})\cdot 0.4 + e^*_{+1} \cdot 0.6 = 0.5
> $, and $P \bigl(f^*(x)=+1\bigr) = 1-P \bigl(f^*(x)=-1\bigr) = 0.5$.
>
> For noisy labels:
> $P \bigl(\tilde{Y}=-1\bigr) = P \bigl(\tilde{Y}=-1|Y=-1\bigr) P(Y=-1) + P \bigl(\tilde{Y}=-1|Y=+1\bigr) P(Y=+1) = (1-e_{-1})\cdot 0.4 + e_{+1} \cdot 0.6 = 0.52
> $, and $P \bigl(f^*(x)=+1\bigr) = 1-P \bigl(f^*(x)=-1\bigr) = 0.48$.
>
> For the joint distribution:
> $P\bigl(f^*(X)=-1,\tilde{Y}=-1\bigr) = P\bigl(f^*(X)=-1,\tilde{Y}=-1|Y=-1\bigr)P(Y=-1) +P\bigl(f^*(X)=-1,\tilde{Y}=-1|Y=+1\bigr)P(Y=+1)
> = (1-e^*_{-1})(1-e_{-1})\cdot 0.4+e^*_{+1} \cdot e_{+1}\cdot 0.6 = 0.296
> $, and $P\bigl(f^*(X)=-1,\tilde{Y}=+1\bigr) = P\bigl(f^*(X)=-1\bigr) - P\bigl(f^*(X)=-1,\tilde{Y}=-1\bigr) = 0.264$.
>
> Further
> $
> P\bigl(f^*(X)=+1,\tilde{Y}=-1\bigr) = P\bigl(\tilde{Y}=-1\bigr) - P\bigl(f^*(X)=-1,\tilde{Y}=-1\bigr) = 0.224,
> $ $~P\bigl(f^*(X)=+1,\tilde{Y}=+1\bigr) = P\bigl(f^*(X)=+1\bigr) - P\bigl(f^*(X)=+1,\tilde{Y}=-1\bigr) = 0.216.
> $
>
> With above, the entries in Delta can be computed easily, for instance
> $$
> \Delta(1,1) =P\bigl(f^*(X)=-1,\tilde{Y}=-1\bigr) - P\bigl(f^*(x)=-1\bigr)\cdot P\bigl(\tilde{Y} = -1\bigr) = 0.296 -  0.5*0.52 = 0.036
> $$
>
> We do notice a miscalculation in the draft; we will update!
>
>
> Lemma 6:
>
> We apologize for the unclear last step - this was due to the reshuffle of results. We have used a partial results in Lemma 1. We updated the draft:
>
> Since $R^*$ can be written as a function of $X$ and $Y$, due to conditional independence between $R$ and $X$ (conditional on $Y$), by chain rule $(R^*=-1, R=+1) = P(Y=+1) (1-e_{+1})e^*_{+1} + P(Y=-1) e_{-1} \cdot (1-e^*_{-1})
> $.
>
> Since
> $$
> P(R=+1) = P(Y=+1) (1-e_{+1})+P(Y=-1)\cdot e_{-1},
> P(R^*=+1)
> = P(Y=+1) (1-e^*_{+1})+P(Y=-1)\cdot e^*_{-1}
> $$
>
> We then have
> $$P(R^*=+1, R=-1)  - P(R^*=+1) P(R=-1)=-P(Y=+1)P(Y=-1)(1-e_{+1}-e_{-1})(1-e^*_{+1}-e^*_{-1}).$$ The above equation establishes the last equivalence.
>
>
> On p:
>
> We accidentally dropped the definition of $p := P(Y=1)$, which is simply the marginal distribution of true label Y.
>
>
> On sensitivity of alpha:
>
> Most of alphas are close to 1. We thank the reviewer for the comment; we will provide the details and discussion in the updated draft.
>
>
> We hope the above clarifies!

---

### Public Comment · ~Nontawat_Charoenphakdee1 · 2019-10-17
**Related work is missing: There are several papers that considered learning from noisy labels without knowing noise rates with theoretical guarantee**

Although the authors suggested this is the first theoretical result of learning with noisy labels without knowing noise rates, I would like to point out that there are already several works. I understand that the submitted work considers class-conditional noise (CCN).
=====
I think the most important work of noisy label learning without knowing noise rates is the use of a symmetric loss, i.e., "\ell(f(x),1)+\ell(f(x),-1) = Constant".

First, I would like to point out the following two important works:

[1] Manwani et al.: Noise tolerance under risk minimization, IEEE Transactions on Cybernetics 43 (2013)
[2] Ghosh et al.: Making risk minimization tolerant to label noise Neurocomputing 160 (2015): 93-107.

In [1] and [2], they suggested that using a symmetric loss, under symmetric label noise (or RCN in the submitted paper), without knowing a noise rate, the minimizer of the expected noisy risk is identical to the minimizer of the clean risk (Thm.1 of [2]).
([1] focused on the 0-1 loss while [2] extended it to symmetric losses (0-1 is symmetric))

Definitely, RCN is a special case of CCN. But [1] and [2] also mentioned that in a more general scenario (i.e., non-uniform noise, which CNN is a special case of it), the robustness result still holds if an additional assumption is satisfied (Thm.2 of [2]).

This work of noise robustness without knowing the noise rate was also extended to the multiclass loss by the following paper:

[3] Ghosh et al.: Robust loss functions under label noise for deep neural networks. AAAI2017.

The following work also discusses about a symmetric loss with a rigorous theoretical guarantee (for RCN).

[4] van Rooyen et al.: Learning with symmetric label noise: The importance of being unhinged, NeurIPS2015

The advantage of the symmetric condition and 0-1 loss is also discussed in a more general noise scenario (mutually contaminated framework (MC), which CCN is a special case) with more evaluation metrics (balanced error rate and AUROC) in the following works:
[5] du Plessis et al.: Clustering Unclustered Data: Unsupervised Binary Labeling of Two Datasets Having Different Class Balances, TAAI2015
[6] van Rooyen et al.: An average classification algorithm. arXiv:1506.01520, 2015
[7] Menon et al.:  Learning from corrupted binary labels via class-probability estimation, ICML2015
[8] Charoenphakdee et al.: On symmetric losses for learning from corrupted labels, ICML2019

[5] suggested that in MC framework (including CCN), if p=0.5, we may estimate density-difference to get a Bayes optimal classifier without knowing the noise rate (not ERM approach). [7] showed that CCN is a special case of MC framework.
[6], [7], [8] emphasized that when the data is balanced (p=0.5), we do not need to know the noise rate to solve this problem. While [7] focused on a loss that can estimate the class posterior probability (e.g., logistic, squared), [8] suggested that using a symmetric loss is advantageous (0-1 loss is symmetric and the submitted paper also discussed about the robustness of 0-1 loss).

In [8], we provided experimental results to validate the importance of symmetric conditions on UCI datasets, CIFAR-10, and MNIST. We can see that even very high noise, we can get 96 percent accuracy in twonorm too.

Also, the following paper is worth mentioning:

[9] Zhang and Sabuncu: Generalized cross entropy loss for training deep neural networks with noisy labels, NeurIPS2018

In [9], they argued that although the symmetric loss for multiclass (MAE) proposed by Ghosh AAAI2017 is robust, it is hard to train for challenging datasets, and they relevate this bottleneck by trying to make it easier to train. This also suggested that using a symmetric loss is not a silver bullet and other approaches may be interesting to explore. All 1-9 (though 9 may be a bit experimental) works have a theoretical guarantee and did not consider the noise rate is given.
=======
In proof of theorem 3 in the submitted paper, the authors used a symmetric property and thus I think the relationship of the submitted work must be highly related to symmetric losses. Remark 2 in theorem 5 may coincide with the robustness of 0-1 loss under RCN [1][2]. Lemma 4 seems related to the robustness of the balanced error rate (BER) in [6][7][8]. I think unweighted peer loss corresponds to minimizing the balanced error rate of the noisy data.

Nevertheless, to the best of my knowledge, the idea of the proposed peer loss functions in this paper is novel and definitely worth considering. It would be great to compare this proposed method with the existing work (e.g., symmetric loss). Thus, I strongly feel that the authors should discuss the related work since it is undoubtedly highly related. And the discussion about the relationship with them was unfortunately missing in the initial submission. I hope my comment helps the authors improve the related work section and the experiments in this paper.

---

> ### Author Response · Authors · 2019-10-18
> **thank you for the pointers! focused more methods for asymmetric noise rates; have experimented with symmetric loss**
>
> Dear Nontawat,
>
> We greatly appreciate your comments. Indeed reading through them helped us refresh our thoughts about the symmetric loss.
>
> We were aware of symmetric loss functions proposed in the literature, when the error rates are symmetric. We indeed implemented one symmetric loss (sigmoid loss) and observed that it performs worse than both peer loss and the surrogate loss proposed by Natarajan et al 2013 (which is one of our benchmark method), for both symmetric and asymmetric error rate setting. The above observation partially motivated us to to focus on the methods proposed for asymmetric error rate and decided to not include everything from the broad learning with noisy data literature. But thank you for reminding us that under some conditions (e.g., L( f(x), 1) + L( f(x), −1) = Constant), the techniques developed for the symmetric case are also robust to asymmetric noises. We aimed for a generic method that would not require much assumption on the error rates and the loss functions. In the next version, we will make it clear that "learning with noisy data without knowing noise rates" has been studied in the symmetric setting, and the asymmetric setting under conditions.
>
> Following the comment, we have just tried replacing the cross-entropy loss in peer loss with the sigmoid loss, but we didn’t observe clear improvement either (but thank you!).
>
> We will add these references to our next version, and we will add symmetric loss to our baseline competitors. This will indeed help us improve our related works and the experiment session to be more comprehensive.
>
> We are keen to read [8] and understand the conditions that would make symmetric loss work and more robust.
>
> In our theorem 3, our symmetricity assumption is mainly in ground truth label’s prior (p=0.5), which we believe is different from the “symmetricity” in the symmetric loss setting. We do agree Lemma 4 is related to [6-8]; will definitely cite. We do appreciate you pointing this out. We will better clarify.
>
> Happy to discuss further!

---

> > ### Public Comment · ~Nontawat_Charoenphakdee1 · 2019-10-19
> > **Thanks for the response! I believe the direction you are going is useful to the community.**
> >
> > Dear authors,
> >
> > I appreciate the author's prompt response. For those who may have read my first comment, I have to clarify that although my comment was perhaps a bit long, the main message is to only ask for adding the discussion about the related work for completeness. So I suggested papers that I think they can be related. Also, let me clarify that I didn't say that the proposed idea has already existed. I think the proposed idea is novel.
> >
> > As the authors suggested, symmetric losses are known for a strong robustness result when the noise is RCN (Ghosh et al.). When it comes to CCN, it needs more assumption to be robust, which that assumption can be unrealistic. Thus, it is still important and relevant to explore the noise robustness in CCN without knowing the noise rate as this paper is doing.
> >
> > Thank you for reporting the experimental result. The potential reason why a symmetric loss may still not perform well under noise is that, although the minimizer of the noisy risk and the clean risk are identical, the sample complexity can be much bigger (by a big constant) as the noise rate increases. That's why having the clean data is better than having the noisy data (this explanation cannot be seen from the analysis of minimizers, but the analysis of the generalization error bound). But I guess in theorem 8 in the peer loss also suggested that the bound is looser when it is noisy ($\frac{1}{1-e_{-1}-e{+1}}$ is large). So another reason might be it is not easy to train compared with the cross-entropy loss.
> >
> > =====
> > About theorem 3, I just saw that the symmetric condition I mentioned is in page 14,
> >
> > "When $\ell$ is the 0-1 loss we have $\ell(f(X), −1) + \ell(f(X), +1) = 1$"
> >
> > This condition gives the theorem, which proved that both the left and the right hand side of the final result have the same minimizer. So I guess Thm.3 also true for all symmetric losses if we did not assume other specific property of 0-1 loss until that point. But I think it's fine not to generalize the result if it causes confusion to the reader.
> >
> > For p=0.5 (the class prior of the target clean risk), it is the situation where the accuracy and the balanced accuracy are identical, thus minimizing balanced error rate can be robust to noise as this paper suggested. Anyway, [8] did not discuss the general scenario but only focused on the balanced error rate and AUROC. Thus, the problem considered in this submitted paper is more general than [8] in this sense.
> > =====
> >
> > potential typo:
> > before the beginning of lemma 2: Sgn(∆) is simply the identify matrix -> identity matrix.
> >
> >
> > Thanks for the nice work!
> > I'm looking forward to the updated version.

---

> > > ### Author Response · Authors · 2019-10-20
> > > **thank you for your clarification and careful reading**
> > >
> > > I see. Right, for 0-1 loss we leveraged the symmetricity property as you correctly pointed out. This now indeed looks generalizable to other losses when the sum is a constant. (we see the connection with the pointed out references now)
> > >
> > > Thanks for pointing out the equivalence between the two measures when p=0.5 - this is helpful for us to understand [8] and our differences.
> > >
> > > We appreciate your advises on why the symmetric losses might not see a good performance. We will play around with it more!
> > >
> > > Best,
> > > Authors

---

### Public Comment · ~Yilun_Xu1 · 2019-11-04
**Important related work is missing**

Hi,

[1] also does not require to know the noise rate, and is provably not sensitive to noise patterns and noise amount. I wonder how the noise-robust function[1] performs in your setting. In experiments of [1], the noise pattern has both symmetry and asymmetry patterns, as well as diagonal-dominant and non-diagonal-dominant patterns.

Thank you!


[1] Yilun Xu, Peng Cao, Yuqing Kong, and Yizhou Wang. L_DMI: A novel information-theoretic loss function for training deep nets robust to label noise.  NeurIPS 2019

---

> ### Author Response · Authors · 2019-11-04
> **thanks for the pointer. Our differences & the pointed out article was too fresh for us to include**
>
> Dear Yilun,
>
> Thank you for the pointer. Indeed we are not aware of the work until the posted comment. We'd like to note that the pointed article [1] was submitted to arXiv on Sep 8, 2019, which was roughly two weeks before ICLR deadline. To be fair, we feel that would be too fresh for us to include a thorough comparison - around the point, we have pretty much finalized our results and were focusing on writing up our draft. But we definitely will include this in our next version; in fact we are keen to observe the difference between the methods.
>
> A (very) quick read seems to imply the following differences:
>
> - We aimed for a simple-to-optimize loss function that can easily adapt to existing ERM solutions. After the submission, we have observed other successes in adapting peer loss to more sophisticated neural network solutions, differentially private ERM, and semi-supervised learning etc. [1] seems to require estimations of a joint distribution matrix, and then to invoke computing a certain information theoretical measure.
>
> - With above concern, it is not entirely clear to us about the sample complexity requirement in [1], and the sensitivity to noises in this estimation. We do provide calibration guarantees and generalization bounds.
>
> - We provide conditions when the loss functions are convex. In general, we do think computationally peer loss functions are easy to optimize with, in comparing to information theoretical measures.
>
> - We provide extensive comparisons with the state-of-the-art learning with noisy label approaches (which even have access to the error rate information).
>
> - We provide theoretical guarantees for both Bayes' optimal classifiers and general hypothesis classes, and establish a broad connection between learning with noisy labels with peer prediction score functions. It is true we have focused on CA, but the connection implies the possibility of applying other peer prediction functions.
>
> Nonetheless, it looks a fun paper to us to read, and we will certainly do.
>
> Happy to discuss further.
>
> Best,
> Authors

---

### Author Response · Authors · 2019-11-12
**Revised draft uploaded; summary of main changes; added experiment results (comparing to other works, multi-class); related works**

Dear reviewers and all,

We have updated our draft according to your comments and uploaded a revision. Major revisions are highlighted in blue. We would like to thank the reviewers and our public comments again for the extremely helpful comments. We won’t be able to reach this revision without your help. We believe our draft has improved significantly. We would highly appreciate it if you could read our revisions and let us know if you have any further concerns.

To summarize our main changes:

1. We have updated our introduction and related work to better position our contributions. We want to emphasize that we have focused on asymmetric error rates – it is true that there exist loss functions which do not require the knowledge of noise rate when the error rates are symmetric (or under specific conditions, e.g., when the optimal Bayes risk is 0). We have added comparisons with symmetric loss functions (we’d like to thank Reviewer 1 and Nontawat for very helpful comments and pointers.)

In response to Reviewer 1 & 2’s comments, we have “toned” down a bit (it was not our intention, but we were not super clear) - we didn’t mean that the estimation of noise rates is not possible (or transition matrix, as used in a number of papers). Our goal is to provide a different and new approach that wouldn’t require this estimation. Our method is useful in settings when the practitioner does not have access to reliable estimates of the noise rates (e.g., when the training data has limited size for the estimation tasks, or when the training data is already collected in a form that makes the estimation hard to perform. This challenge was particularly cited in the CVPR'17 paper pointed out by Reviewer 1 (https://arxiv.org/abs/1609.03683):

"The quality of noise estimation is a key factor for obtaining robustness ..... where estimation destroys most of the gain
from loss correction. We believe that the mix of high noise and limited number of images per class (500) is detrimental
to the estimator. "

2. Compare to a recent NeurIPS 19 paper (DMI): per the request by both Reviewer 2 and a public comment, we have added a discussion about our difference with a very recent work on an information theoretical measure (DMI) that does not require specification of noise rates either. Besides peer loss’ additional theoretical guarantees of generalization and calibration, peer loss has shown consistent empirical advantages in comparing to DMI (we’d like to note that DMI performs fairly well too; in the previous paper DMI was only tested with simple and sparse noise setting. So we provide a more extensive study of it).

3. We further clarified our contributions. Regarding Reviewer 1’s comment on our novelty in light of a CVPR’17 paper, we clarify that the proposed loss correction approach therein requires the knowledge of a noise transition matrix T. Peer loss does *not* require either the knowledge of T nor the estimation of T. The loss correction approach was derived from the surrogate loss function methods (e.g., [Natarajan et al 13]) where the goal is also to define unbiased estimators of the true losses. We have compared with surrogate unbiased loss approach in our paper. We hope this clarifies and justifies our novelty.

4. We added back some preliminary results on multi-class classification (on CIFAR-10 dataset) with an implementation of peer loss in ResNet (replacing cross entropy loss). As we mentioned earlier in the paper, because of that peer loss does not require the knowledge of transition matrix of noise labels, our method generalizes to multi-class labels easily. We added some preliminary analysis for multi-class in Section 4.1 and Appendix. We do observe peer loss is effective compared to DMI and another baseline. This implementation also demonstrates the usefulness of peer loss in a deep learning task.

5. Notations are updated to avoid confusion.

Best,
Authors

---

> ### Author Response · Authors · 2019-11-15
> **updated organization and presentation**
>
> Updated the organization and  presentation, as suggested by Reviewer 2.

---

### Decision · Program_Chairs · 2019-12-19

**Decision:**

Reject

**Comment:**

Thank you very much for the detailed feedback to the reviewers, which helped us better understand your paper.
Thanks also for revising the manuscript significantly; many parts were indeed revised.
However, due to the major revision, we find more points to be further discussed, which requires another round of reviews/rebuttals.
For this reason, we decided not to accept this paper.
We hope that the reviewers' comments are useful for improving the paper for potential future publication.